# Exploring mobility patterns and social health of older Canadians living at home to inform decision aids about housing: A mixed-methods study

Diogo Mochcovitch[1], Allyson Jones[2], Joshua Goutte[1,3], Karine V. Plourde[1], Roberta de Carvalho Corôa[1,4], Marie Elf[5], Louise Meijering[6], Jodi Sturge[6,7], Pierre Bérubé[8], Stéphane Roche[3,9], Sabrina Guay-Bélanger[1], France Légaré[1,4]*

**1** VITAM—Centre de Recherche en Santé Durable, Centre Intégré Universitaire de Santé et de Services Sociaux de la Capitale-Nationale, Quebec, Quebec, Canada, **2** Department of Physical Therapy, Faculty of Rehabilitation Medicine, Physical Therapy, University of Alberta, Edmonton, Alberta, Canada, **3** Département des sciences géomatiques, Faculté de foresterie, de géographie et de géomatique, Université Laval, Quebec, Quebec, Canada, **4** Department of Family Medicine and Emergency Medicine, Faculty of Medicine, Université Laval, Quebec, Quebec, Canada, **5** Department of Nursing, School of Health and Welfare, Dalarna University, Fallun, Sweden, **6** Population Research Center, Urban and Regional Studies Institute, University of Groningen, Groningen, The Netherlands, **7** Department of Design, Production and Management, Faculty of Engineering Technology, University of Twente, Enschede, The Netherlands, **8** Greybox Solutions, Montreal, Quebec, Canada, **9** Institut EDS—Institut en environnement, développement et société, Université Laval, Quebec, Quebec, Canada

* france.legare@fmed.ulaval.ca

## Abstract

### Introduction

Many tools support housing decisions for older adults but often overlook mobility patterns and social health. We explored these factors in older Canadians living at home to inform housing decisions.

### Methods

We conducted a mixed-methods study with 20 older adults (65+) from Quebec and Alberta living independently or in senior residences with outdoor mobility. Data collection included sociodemographic information, GPS tracking, walking interviews, daily journals, and in-depth interviews. Data from interviews, which explored physical and social assets and barriers to social health and mobility, were analyzed using deductive content analysis in NVivo 12. GPS data were subjected to spatial analysis in QGIS (Quantum Geographic Information System) to map activity spaces and mobility patterns by the number and distance of activities, activity types, and modes of transportation. Daily journals were transcribed into an Excel spreadsheet and compared with GPS data. Overall analysis was guided hierarchically by qualitative data, utilizing verbatim narratives and visualization (activity space maps) to illustrate data convergence.

**Data availability statement:** As data were collected and stored, they were all de-identified using anonymous identifiers to ensure confidentiality. However, due to the nature of GPS data, there remains a risk of potential re-identification within raw datasets. The Ethics Committee approved the collection and analysis of the data only for the specific study and not for other purposes. The Ethics Committee requires that the data collected remains securely stored and not be shared. Researchers interested in accessing this data may submit a request to the Comité d'éthique de la recherche du CIUSSS de la Capitale Nationale at Simon Trempe at 555 boul. Wilfrid-Hamel, bureau E-115, Québec (Québec), G1M 3X7, or via email at simon.trempe.ciussscn@ssss.gouv.qc.ca (more information on the ethics committee is available at https://www.ciusss-capitalenationale.gouv.qc.ca/mission-universitaire/recherche/ethique-recherche/sante-population-premiere-ligne). We report all data relevant to answer our research question in the paper.

**Funding:** This publication is part of the COORDINATEs study (project number: 9003037412), which is funded by the Joint Programming Initiative More Years, Better Lives, represented by the Canadian Institutes of Health Research (CIHR). Grant details: Acronym of the collaborative project: COORDINATEs Full title of the project: teChnology tO suppORt DecIsioN Making about Aging aT homE Grant number awarded: #155230 Initials of the author who received the funding: FL The sponsor did not play any role in the study design, data collection and analysis, decision to publish, or preparation of the manuscript.

**Competing interests:** The authors have declared that no competing interests exist.

## Results

Among 20 participants, 14 completed all activities, including GPS trackers. GPS maps showed participants mostly left home to drive for shopping or walking. Over 14 days, participants made an average of 10.4 (±5.8) trips and traveled 186.9 km (±130.4), averaging 16.8 km (±29.8) per day. Transportation modes included car (n=9), walking (n=5), and bus (n=2). Daily journals revealed that participants typically traveled alone. Interviews identified physical assets as libraries and supermarkets (n=10), while social assets were family support when desired (n=13) neighborhood familiarity (n=14), both contributing to social health. Winter weather was the most cited mobility barrier (n=13).

## Conclusions

These findings provide actionable insights to guide the development of user-informed decision support tools tailored to the housing decisions of Canadian older adults.

## Introduction

With the increase in life expectancy, challenges related to the health, well-being and housing of older adults arise. Among problems faced in the later phase of life, loss of autonomy, feelings of insecurity, and frailty most frequently impact well-being [1]. Loss of autonomy notably hinders the performance of daily tasks [2]. These factors including their social isolation can affect the social health of older adults [3].

When older adults lose autonomy, they face various health-related decisions. In spite of many fitness programs developed for older adults to increase physical activity and improve their physical function, loss of autonomy is associated with physiological aging [4].

A cross-sectional survey conducted with a pan-Canadian web-based panel of older adults aged 65 and over, assessed clinically significant decisional conflict (CSDC) [5]. The survey revealed that housing decisions – specifically, the decision-making process regarding whether to age at home or move into institutional residential care – are the most commonly reported difficult decisions faced by older adults. This finding was confirmed by studies among caregivers of older adults [6] and their home care providers [7].

Studies show a strong relationship between wellbeing and social participation among older adults [8]. Other studies suggest that moving home may remove these social networks and other informal support systems [9]. One review of support for older adults making housing decisions suggests that such support is generally undermined by lack of attention to the whole person and lack of preparation for the move [10]. Secondly, while there are studies showing that mobility among older adults is important to a sense of autonomy and social engagement [11–14], few studies have explored the relationship between mobility and social health in the context of older adults' housing decisions [9,15]. As an example, an older adult who is used to visiting nearby community centers may be relocated to an area with limited access to public transportation, thereby restricting their social interactions and undermining their social health. If tools that support housing decisions, such as decision aids, fail to take mobility as a social asset into account, increased decision regret is likely [16].

A decision aid is a tool that supports informed value congruent decision making by guiding people to evaluate the evidence on the benefits and limitations of different options based on personal values and priorities before making a decision [17]. More often than not, decision aids to support housing decisions focus exclusively on the older adults' health condition and on their physical safety [18].

Many housing decisions are about moving an older adult into institutional residential care. If physical safety and nursing care are the principal considerations in this decision, an institutional setting may be regarded as the best choice. However, most older Canadians wish to grow old in their own homes for reasons other than their physical health and safety [10]. Without considering these additional reasons, loss and disappointment may follow their decision.

The social health of older adults is as important as their physical health to their overall well-being [8,19]. Indeed, the influence of social relationships on the risk of death are comparable with well-established risk factors for mortality such as smoking and alcohol consumption [20]. There is a growing interest in expanding the focus of health to include social health [21,22]. Social health emphasizes the individual's capacity to engage in social activities, maintain a sense of purpose and fulfillment in life, and retain some degree of independence (e.g., going for a walk alone) [22]. Social health entails sustaining meaningful relationships, social participation, functioning to the full extent of one's potential, and coping with the practical consequences of aging. For older adults, these capacities contribute to healthy aging [23].

Mobility plays a key role in social health. Although the definition of mobility refers solely to the physical ability to move independently from one place to another [24,25], there are influencing factors that shape the underlying purpose of this movement (e.g., social interactions, family engagements, work, and leisure activities) [26]. In this study, we consider mobility as a physical phenomenon, but also the influencing factors that determine its purpose. A related concept in gerontological research relevant to housing decisions is understanding how older adults interact with their existing physical and social environments, how this interaction changes with time, and how this impacts their social health [27]. Some studies have used the idea of "lifespace mobility," or the range of physical environments in which older adults move during a specified time period, to measure mobility and its determinants [28–31]. The notion of "activity space," which also combines these notions of space and time, is the collection of places outside a person's home (but not necessarily within their own neighborhoods) that they routinely or occasionally visit, and the corresponding travel routes that they use to get there [22,32,33]. The concept has been widely used in studies investigating the social determinants of health, access to health care, diet, physical activity and social inequalities [34–36]. A few studies have explored activity spaces in the lives of older adults [22,35–38], but none in the context of developing decision aids for making housing choices.

While healthcare professionals and family caregivers involved in housing decisions with older adults may be conscious of their growing health burdens and safety needs [6,7], they may be less aware of these existing social and physical assets and older adults' access to them, which should play an equally important role in the housing decision [39–41]. For instance, it is important to identify the social supports, such as friends and family, that currently help maintain an older adult's social health [42]. Or volunteering in their community may provide them with a healthy sense of self-worth, and their mobility may make this possible [43]. Therefore, it is important to understand the motivations that help them remain mobile and maintain their physical health, and to consider whether, if they decide to move to a different home or neighborhood, these social and physical assets will still be available to them.

Based on these gerontological concepts, we sought a new perspective on how older adults' mobility patterns, i.e., the quantified mobility of outdoor travels [26], and activity spaces affect their social health. Following earlier exploratory studies [22,44,45], we hypothesized that measuring the mobility patterns of older adults, in combination with qualitative interviews and journals regarding the social advantages that are associated with that mobility, would provide a portrait of their existing social and physical assets that constitutes essential information for their housing decisions. Existing research often isolates these factors [46–48] or relies on

self-reported data [49–51]. Therefore, with the overall goal of informing the design of decision aids, we aimed to map the activity spaces of older adults in two Canadian provinces by combining GPS data with narrative sources (interviews).

## Methods

### Study design and context

This mixed-methods study is part of a larger international initiative, teChnology tO suppORt DecIsioN Making about Aging aT homE (COORDINATEs), an interdisciplinary multi-pronged research program conducted in Canada, Sweden and the Netherlands to understand the mobility patterns and experiences of older adults living at home and how this data can improve autonomy and inform shared-decision making about housing options [45]. As this research involves a variety of different data sources, we opted for a mixed methods approach guided by the Good Reporting of A Mixed Methods Study (GRAMMS) guideline [52]. At the same time, our study design can be better described as a convergent mixed-methods approach [53] whereby the different methods are integrated through building on one data source after another [54]. In this method, the results from GPS data are used together to inform our understanding of other qualitative data sources.

We took a matching approach to converging data [55], involving intentionally designing our data collection instruments to have related items so that all instruments would elucidate data about the same issues of mobility and social health. In terms of the methodological dimension of our data merging, we took a qualitatively driven approach, i.e., guided hierarchically by the qualitative data. Concretely, we used the technique of matching data from qualitative questions with the quantitative GPS data, integrating our data at the interpretation and reporting level using both narrative (the discussion) and visualization (the activity space maps) to illustrate where the databases converged, complemented, conflicted, or diverged.

### Ethical approval

Ethical approvals were obtained from the Health Ethics Research Board at the University of Alberta (Pro00087478) for Alberta and from the Integrated University Health and Social Services Centres (CIUSSS) de la Capitale Nationale (2019–1519) for Quebec. Written and informed consent was obtained and secured from all participants prior to their enrollment in the study.

### Participants

The Canadian sub-study recruited participants from two provinces, Alberta, and Quebec, using a variety of outreach methods. The screening took place in a fall prevention clinic [56], through community engagement, and by distributing flyers and posting them in seniors' residences. Eligibility criteria included: a) aged 65 years or older, b) living autonomously at home or in a seniors' residence, c) independent outdoor mobility.

### Data collection

While much data on older adults is gathered from staff members in residential settings or family members, we wished to capture the direct experience of older adults themselves, as this would empower them to discuss their current and future needs [57,58]. A combined approach to gathering data improves data accuracy by cross-validating information through multiple sources [53,59]. Data were gathered using five distinct methods: i) a sociodemographic and other self-reported information, ii) a walking interview iii) GPS tracking, iv) daily journal

entries, and v) an in-depth interview. The data were collected at different times in each province. In Quebec, data collection took place between August 10, 2019, and March 5, 2020. In Alberta, it began on February 1, 2020, but was interrupted due to the COVID-19 public health emergency. It resumed between September 29, 2021, and February 28, 2022.

**Sociodemographic and other self-reported information.** Upon meeting the inclusion criteria and providing written consent, participants completed the survey, which was comprised of sociodemographic information, perception of self-reported health status, assistance required for daily tasks, and technology use. For self-reported health status, we used a 5-point Likert scale [60], allowing participants to rate their health based on subjective experience, with options ranging from poor, fair, good, very good, to excellent. For questions regarding assistance required with daily tasks, we based our items on the Instrumental Activities of Daily Living (IADL) Scale, which assesses the ability to perform complex everyday tasks (e.g., managing finances and medications) evaluating whether the participants receive or provide help with those tasks [61]. Regarding technology use, we asked participants to specify the types of technology they typically use in their daily life.

**GPS.** Each participant was shown how to use a GPS tracking device (QStarz BT-1000X) and for over 14 days they attached the devices around their waists when they left the house. A minimum of 14 days is recommended to accurately record a person's routine activity spaces [62]. We used GPS tracking to provide an objective measure of physical distance and type of participants' activity spaces. GPS observations [63] also allowed greater precision of time spent in particular areas or at specific activities rather than capturing only particular locations. Finally, GPS tracking captured locations in smaller time intervals (e.g.,1 or 5 seconds intervals), and had a longer battery life than smartphones which meant less data was lost.

**Daily journal.** The daily journal was a paper-based matrix which participants completed over the 14 days, detailing their activities outside their home. Each daily entry included information such as purpose of the activity, mode of transportation, companions or absence thereof, and if they needed to use a mobility aid. The journal also complemented the GPS data with information about what the people or places visited meant to them.

**Walking and debriefing interviews.** Participants were interviewed by a research assistant while they were walking in familiar places and routes, typically close to their homes. The purpose of walking interviews was to capture details about participants' activity spaces on a moment-by-moment basis. Participants were asked about the physical environment (street quality, sidewalk quality, problems, or need to change their routes) and social environment (e.g., the people they would meet, visits with family, friends, etc.). These interviews typically took 15 minutes with some participants taking rest breaks during the walk.

Another interview session was scheduled with participants after the 14-day period at a mutually convenient time for a thorough in-depth interview which generally lasted 45 to 60 minutes. This interview was another opportunity, through prompts in their daily journals, to offer further insights into their overall experience.

## Data analysis

The goal of our data analysis was to match qualitative data with quantitative data, integrating or merged data at the interpretation and reporting level using both narrative and visualization (activity space maps) to illustrate where the databases converged, complemented, conflicted, or diverged.

We conducted this analysis in a hierarchical manner, layering one source upon another, with GPS data and interviews serving as primary sources, complemented by daily journal entries. We chose this approach due to the building nature of the data [54], where the

interviews deepened our understanding of the mobility captured by the GPS tracking, uncovering the factors influencing the mobility. For instance, GPS data may reveal a participant's choice to take a longer route to the supermarket instead of a shorter one, while interview data could indicate that this choice serves as a coping strategy to avoid a noisy avenue that causes stress. The combination of methods allowed us to produce a comprehensive analysis of social health in the mobility patterns of older adults [53]. For participants who did not provide all primary data sources (e.g., those who did not use the GPS properly but provided interview data), their data were still analyzed but only in the relevant sections (e.g., their responses were included in the qualitative analysis and categorized under the appropriate themes and codes).

Upon completion of data collection, interviews were transcribed, and data anonymized (manual encryption of personally identifiable information). A deductive content analysis was performed on the transcripts based on an adapted version of the Asset-Based Approach to Community Development (ABCD) framework [44,64] using NVivo 12. Qualitative and quantitative data (socio-demographic, self-reported health and ability to perform daily tasks) were then organized into a descriptive summary. Statistical Analysis System (SAS) software was used to describe characteristics of participants.

For analysis of the GPS data, cleaning of tracks was done to eliminate the congregation of points around the households of the participants. Where the information regarding participants homes' and where they went (based on diary data, see below) was already known so we were not omitting any information. While there were instances of data points being recorded erroneously, there was a distinct pattern in which the errors appeared, and it was obvious that these were not feasible patterns for any participants (two points at different locations at the same time). The data was sorted by time and then traced from one point to another, and the only points discarded were the congregations of points when a participant was not moving. The main purpose of the GPS analysis was to illustrate the activity space of participants, so the cleaning process was done for illustrative purposes only and no track was omitted due to any undesirable outcomes by any of the authors.

**GPS and daily journal.** GPS location data (distance travelled, activity type and duration) was used to track participants' mobility. Relationships between mode of transport and distance were evaluated and key statistical indicators (mean/ standard deviation) of the distance in each mode of transport were used to create activity space maps.

The following indicators were used to assess activity spaces, and the social and physical abilities of the participants:

*Number and distance of activities***:** We used QGIS software to determine the distance (kilometers) of the trips and average distance travelled per day. QGIS is a free, open-source cross-platform geographic information system application that helps view, edit, print, and analyze geospatial data [65]. The purpose was to understand participants' physical ability and the distance travelled from home. The number of trips per person was also measured over the 14 days.

*Activity type***:** We examined the stops on participants' trips using QGIS software and assigned them to a specific activity using a Google Maps base map, such as restaurants, stores, pharmacies, medical clinics or hospitals, sports centers, cultural centers, and grocery stores. The type of activity chosen was also a sociability indicator such as going to a bingo hall which has greater potential for social interactions than going to a pharmacy.

*Mode of transportation***:** The type of transportation was analyzing using GPS data along with the QGIS software. To distinguish whether participants were walking, taking the bus, or driving, we relied on the speed data point as a key indicator, given the substantial speed variations between these activities.

Finally, types of activity were correlated with distances travelled and modes of transportation.

The daily journals were our second data source for information about distances travelled and modes of transportation. They were collected, digitized, and subsequently transcribed into Excel spreadsheets. This information in Excel was compared with GPS data to confirm or supplement it (e.g., destination, purpose of activity, mode of transportation).

**Walking interviews and in-depth interviews.** Given that the debriefing interview centered on themes explored during the walking interview, which in turn served as a corroborative cross-check of the daily journal entries, we decided to merge the walking and debriefing interviews to gain a comprehensive understanding of the interviews and their underlying themes. Interviews were transcribed verbatim by independent researchers. The resulting transcriptions were then imported into NVivo 12 software, facilitating the qualitative data analysis process. Employing an approach based on a priori codebook from our previous work [44] and adapted to the Canadian context, two independent researchers (KP and DM) conducted a deductive content analysis [66] with 7 overarching themes: physical and social assets (e.g., benches, family members), neighborhood descriptions (e.g., services, stores), mobility descriptions (e.g., with whom, weather, means of transportation), approaches to mobility challenges (e.g., adapt, avoid, cancel), navigation strategies (e.g., points of reference, asking for help), and the impact of Covid-19 on mobility patterns. We also coded reactions to the data collection methods (GPS and daily journal), a seventh theme. We coded and cross-referenced themes to derive meaningful insights. To ensure the integrity of the analysis, the researchers engaged in consensus-building discussions during codification to reach the same understanding of code definitions. After this first consensus, we sought input from an experienced third-party qualitative researcher (RC) to assure consensus. A supplementary secondary analysis [67,68] of the most prevalent codes were performed to establish broader categories. This allowed us to facilitate the interpretation and narrative presentation of qualitative results in relation to the quantitative data sources. Four broader cross-sectional categories relating to social health and mobility were generated: Physical and social assets, social and physical obstacles, adaptation strategies, and the impact of Covid-19. Frequencies of all coding themes are presented in S1 Appendix.

**Illustrative activity space maps.** To illustrate the multi-dimensional nature of participants' mobility patterns and social health, we purposively selected a heterogenous group of 4 participants, 2 from Quebec and 2 from Alberta. For each participant we combined their data [69,70] (i.e., GPS data, diary entries and interview responses) to create personal activity space maps. These maps were generated using QGIS software, with basemaps accessed via the Quick Map Services plugin [71] and sourced from ESRI's Light Gray Canvas basemap [72]. Our choice of the 4 participants for these illustrative activity space maps was driven by our search for the most contrasting mobility patterns, activity spaces, modes of transportation and social health, allowing us to highlight differences between participants and provinces.

## Results

### Flow and characteristics of participants

Of the 25 people approached; 20 participants agreed to participate in the study. Five individuals from Alberta opted out of participating in GPS tracker usage, while one individual died during the research, leaving data only in the form of the responses to the questionnaire. As a result, 14 individuals agreed to utilize GPS devices to map out their activity spaces. Five participants (4 from Alberta and 1 from Quebec) encountered difficulties with the device, especially in charging it, and were unable to record GPS data. Ultimately, 9 participants used

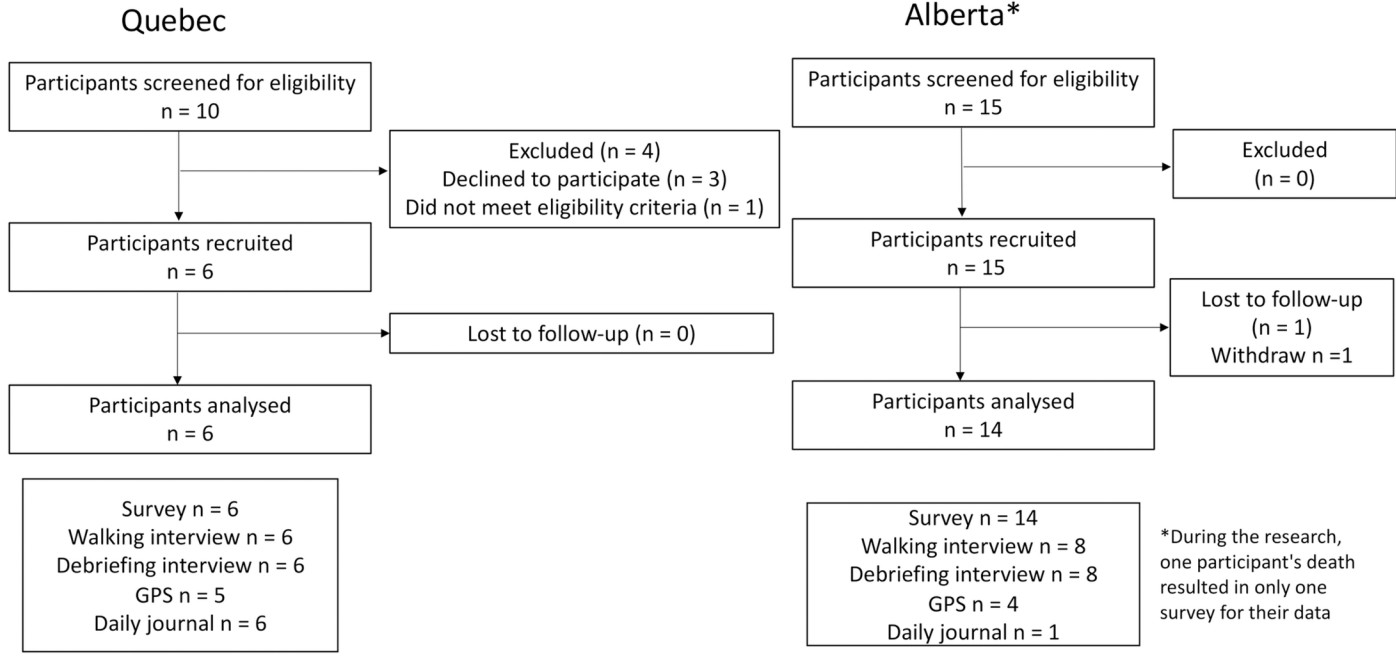

**Fig 1. Participants' flowchart.**

the GPS appropriately. Of the total 14 participants who utilized GPS (both appropriately and inappropriately), 7 also successfully completed the daily journal (1 from Alberta, 6 from Quebec). Fig 1 shows the participants' flowchart.

Table 1 shows details of responses about sociodemographic, self-reported health, assistance with daily tasks, and use of technology. In Alberta, 86% were women (n=12) and their ages ranged from 67 to 89 years old and in Quebec 33% were women (n=2) and their ages ranged from 62 to 90. Overall, they had lived in their current neighborhoods for an average of 15 (±7.9) years. Most were married (n=6) or widowed (n=6). Nearly half had completed higher education (n=9). Nearly half were living with others but independent (n=9), 40% were living alone and independent, and 15% were in a senior residence. "Senior residences" are more akin to adult-only buildings and focus on accommodating autonomous older people (whereas to qualify for public long-term care, residents have been hospitalized and can no longer return home) [73–75]. Half reported they were in "good" health (n=10). In terms of access to technology, 85% used the Internet and 40% used social media while 60% used a smartphone and 60% used a tablet.

## Mobility description: Activity length, type, and means of transport

Of the 9 participants who provided GPS data (5 from Quebec and 4 from Alberta), we generated 9 maps describing their activity spaces. Overall, the mean number of trips per person over the 14 days was 10.4 (±5.8). The average distance travelled per person was 186.9 km (±130.4), and average distance per day was 16.8km (±29.8). Mean distance travelled regardless of the type of transportation was 125 km (±75.4) in Alberta and 236km (±151.2) in Quebec.

In terms of the participants' activity spaces, we divided them into 6 categories over the 14 days of GPS data collection: stores, healthcare facilities, pharmacies, socially focused visits (e.g., children or friends), cultural events, and sports and leisure activities (e.g., take a walk,

**Table 1. Characteristics of participants.**

| Variable | n=20 | % |
|---|---|---|
| **Year of the interview** | | |
| 2019 | 1 | 5.0 |
| 2020 | 12 | 60.0 |
| 2021 | 6 | 30.0 |
| 2022 | 1 | 5.0 |
| **Sex** | | |
| Female | 14 | 70.0 |
| Male | 6 | 30.0 |
| **Marital status** | | |
| Married | 6 | 30.0 |
| Widow | 6 | 30.0 |
| Divorced | 4 | 20.0 |
| Single | 3 | 15.0 |
| Common-law | 1 | 5.0 |
| **Number of children** | | |
| 0 | 3 | 15.0 |
| 1 | 1 | 5.0 |
| 2 | 8 | 40.0 |
| 3 | 3 | 15.0 |
| 4 | 3 | 15.0 |
| 5 | 2 | 10.0 |
| **Location** | | |
| Alberta | 14 | 70.0 |
| Quebec | 6 | 30.0 |
| **Town or village they live in** | | |
| Calgary | 13 | 65.0 |
| Québec | 5 | 25.0 |
| Airdrie | 1 | 5.0 |
| Drummondville | 1 | 5.0 |
| **Education** | | |
| University or higher education complete | 9 | 45.0 |
| High school | 5 | 25.0 |
| Vocational or technical | 3 | 15.0 |
| Elementary | 2 | 10.0 |
| Missing | 1 | 5.0 |
| **Income** | | |
| Pension and reg pension | 10 | 50.0 |
| Pension. reg and additional income | 7 | 35.0 |
| Old age pension | 2 | 10.0 |
| Missing | 1 | 5.0 |
| **Current housing situation** | | |
| Independent. living together with others | 9 | 45.0 |
| Independent. alone | 8 | 40.0 |
| Senior residence | 3 | 15.0 |
| **Individuals benefiting from support with their daily tasks** | | |
| No | 13 | 65.0 |
| Yes | 7 | 35.0 |

*(Continued)*

**Table 1.** (Continued)

| Variable | n=20 | % |
|---|---|---|
| **Individuals who provide help with the daily tasks of others** | | |
| No | 16 | 80.0 |
| Yes | 4 | 20.0 |
| **Health** | | |
| Poor | 1 | 5.0 |
| Fair | 2 | 10.0 |
| Good | 10 | 50.0 |
| Very good | 5 | 25.0 |
| Excellent | 2 | 10.0 |
| **Technology** | | |
| **Type of use** | | |
| **Internet** | | |
| Yes | 17 | 85.0 |
| No | 2 | 10.0 |
| Missing | 1 | 5.0 |
| **Social media** | | |
| No | 10 | 50.0 |
| Yes | 8 | 40.0 |
| Missing | 2 | 10.0 |
| **Smartphone** | | |
| Yes | 12 | 60.0 |
| No | 6 | 30.0 |
| Missing | 2 | 10.0 |
| **Tablet** | | |
| Yes | 12 | 60.0 |
| No | 6 | 30.0 |
| Missing | 2 | 10.0 |
| **Other** | | |
| No | 15 | 75.0 |
| Yes | 4 | 20.0 |
| Missing | 1 | 5.0 |

| Variable | n=20 | Average | Minimum | Maximum |
|---|---|---|---|---|
| Age | 20 | 80.4 | 62.9 | 90.6 |
| Years living in the neighborhood | 20 | 15.6 | 0.5 | 61.0 |

go to a park). For all participants in both Alberta and Quebec, the most frequent purpose of leaving home was to go to a store.

Transportation was categorized data into 3 groups: walking, using a car, and taking a bus. Most participants used more than one type of transportation. Eight participants used a car as their main means of transportation, while 1 participant relied solely on the bus. In Alberta, 1 participant used a car and walking in the same trip. In Quebec, among the 5 participants who used the GPS, 4 used walking as one of their means of transportation. Although residents in both cities relied heavily on cars for transportation, more people in Quebec used public transport than in Calgary, despite Calgary being almost twice the physical size of Quebec [76]. Table 2 provides a detailed overview of the mobility description.

**Table 2. Means of transportation and activity spaces.**

| Province | Partici-pants Ids | Means of transport (Distance traveled in Km in 14 days) | | | | Activity Spaces (Places visited in 14 days) | | | | | | | |
|---|---|---|---|---|---|---|---|---|---|---|---|---|---|
| | | Walk | Car | Bus | Total | Stores | Health Care | Phar-macy | Visit-ing | Cultural Events | Sport and Leisure | Other** | Total |
| Alberta (AB) | 103 | 0 | 108 | 0 | 108 | 5 | 0 | 0 | 0 | 0 | 0 | 0 | 5 |
| | 106 | 0 | 114.6 | 0 | 114.6 | 4 | 1 | 1 | 0 | 0 | 0 | 1 | 7 |
| | 111 | 27.6 | 201.4* | 0 | 229 | 7 | 0 | 0 | 1 | 1 | 0 | 1 | 10 |
| | 113 | 0 | 48.5 | 0 | 48.5 | 3 | 0 | 1 | 0 | 0 | 0 | 0 | 4 |
| Quebec (QC) | 1 | 23.2 | 87.9 | 0 | 111.1 | 4 | 0 | 1 | 1 | 0 | 0 | 1 | 7 |
| | 2 | 0 | 484.4 | 0 | 484.4 | 7 | 1 | 1 | 0 | 0 | 0 | 5 | 14 |
| | 3 | 3.6 | 113.5* | 0 | 117.1 | 6 | 0 | 0 | 1 | 1 | 1 | 1 | 10 |
| | 4 | 9.2 | 201.2 | 17.2 | 227.6 | 11 | 3 | 3 | 2 | 0 | 1 | 3 | 23 |
| | 5 | 6 | 6.1 | 229.7 | 241.8 | 11 | 0 | 0 | 1 | 2 | 0 | 0 | 14 |
| Average in AB and QC together (±SD) | | 7.7 (±10.5) | 151.73 (±139,7) | 27.4 (±76.0) | 186.9 (±130.4) | 6.4 (±2.9) | 0.5 (±1.0) | 0.7 (±0.9) | 0.6 (±0.7) | 0.4 (±0.7) | 0.2 (±0.4) | 1.3 (±1.6) | 10.4 (±5.8) |

*This was made combining car and walking together

** For example, community center, gas station.

**Activity types in relation to distances and modes of transport.** For 1 participant in Alberta, stores were his only activity spaces, and he exclusively used the car. However, the most active Alberta participant had 10 activity spaces and used both walking and car. The 5 Quebec participants also primarily visited stores, but they had more than one activity space, two of which related to sports and leisure (e.g., parks). The most active participant had a total of 23 activity spaces and used walking, car, and bus to go out. Participants' means of transportation, distance and activity spaces are presented in Table 2.

Daily journals were completed by 7 participants (n=1 Alberta and n=6 Quebec). In additional to corroborating GPS data, 6 mentioned that they typically travelled alone and 4 were occasionally accompanied by a family member. Some mentioned they went out "just to get some air." Three mentioned being accompanied by friends, while one participant did not undertake any trips during this COVID period and simply followed their regular exercise routines at their seniors' residence.

## Physical and social assets, social and physical obstacles, adaptation strategies, and the impact of Covid-19

**Physical and social assets.** Of the 14 participants interviewed, most of the social and physical assets mentioned were related to walking. During the walking interviews, most participants (n=10) emphasized the importance of their local institutions (such as libraries and supermarkets) as essential physical assets that helped them stay active. They also mentioned the importance of benches or shelters where they could stop to rest (n=10): "I get tired. I walk in the park and the minute I see a bench I sit down for two or three minutes, then I go on" (P3). One participant organized her route according to the available resting places: "The reason I take this walk is because I have many places where I can stop the walk as soon as I want to" (P103). The design and layout of walking routes also significantly impacted mobility choices (n=13): "There are some excellent walks in this area. Like leaving the main entrance if you walk around, there's a path down here" (P113). Others mentioned the importance of support equipment for walking in winter, such as crampons, walkers, and railings.

The majority (n=13) demonstrated an appreciation of easy walking access to services: "It's all well-organized. There's a pharmacy, a library here, and another library over there" (P5). The importance of familiarity of one's neighborhood was also highlighted by comments from 3 participants on how moving had or would in the future result in loss of orientation and loss of activity options.

In the context of participants' social and emotional responses towards their neighborhood, all 14 participants demonstrated a deep sense of attachment, encompassing not only their geographic surroundings but also the interpersonal relationships with their neighbors. One participant vividly expressed this emotional connection by stating, "I love this neighborhood. I have nice neighbors. I can walk to many places, and I can take the bus downtown. I love this neighborhood" (P111). Familiar places can also stimulate happy memories: "The water was high… It reminded me of being in the Gaspé when I was nine years old" (P3). Walks were often highly social as well as physical activities: "I meet people who are out walking like me. And we talk about the temperature, the nice weather, and about our illnesses" (P3). Dogs turned out to be a physical asset as a motivator go out walking. They are also social assets: "I say all the time that I go out for the sake of Henri [the dog] – but really, it's for me… It's 'good morning, madame, good morning, madame'" (P3).

The main social asset identified by participants was family and neighbors (n=13). Indeed, participants mentioned that family members were their main companions (n=11), mostly their partner and/or children: "When I go to my doctor's appointments… it's my daughter who comes because she's too afraid I'll forget things" (P6). Proximity to family or neighbors (n=13) emerged as a major social asset, although sometimes participants were ambivalent about this: "My daughter drops by every now and again, but I never know when she's going to arrive" (P104).

Three participants, even some in their eighties, mentioned volunteer activities as another social asset: helping out at the hospital, library or even just going walking with fellow volunteers. Participants also mentioned the significance of driving to maintaining their independence. As one participant expressed it, "I'm going to keep driving until I'm 80, and I'm just turning 78 now" (P106). Walking is similarly important for self-esteem and independence: "I complete my entire walk, heading down to the turning point when I go around, and I complete the loop back to my house, if I go for the full route" (P103).

**Social and physical obstacles.** During the winter months, mobility changed for the most of older adults (n=13) both in Alberta and Quebec "When it's cold I don't go out, I just stay home, so there we go" (P104). Both snow and freezing rain were frequently mentioned as mobility hazards. One participant missed gardening in the winter, and others mentioned missing their friends or relatives: "I have to wait until spring to go to Chicoutimi to visit my cousin" (P2). Putting on winter clothing was also a discouragement to going out. "It's not motivating, first you have to put on huge boots, pants, and even if I had the special pants, you also need a big coat, a vest… ah no" (P3). In many locations the city did not clear the sidewalks and participants reported having to walk on the street: "The city doesn't … do the roads here, because of course it's not an important road" (P103).

Certain participants indicated a requirement for specialized mobility aids, particularly canes (n=6): "I have a walker and a cane, and I also have two sticks, so depending on what I'm doing, I alternate with some of them" (Participant 110). Most participants (n=11) disclosed that they had to adapt their routes due to mobility impediments. One even had mobility difficulties accessing her own seniors' residence: "What I do is… just go down to the arcade and then take a stairway up and do it that way. It's not really handicap friendly" (P106).

Older adult participants also mentioned losing friends due to their loss of autonomy, loss of hearing, the relocation of their friends, or their death.

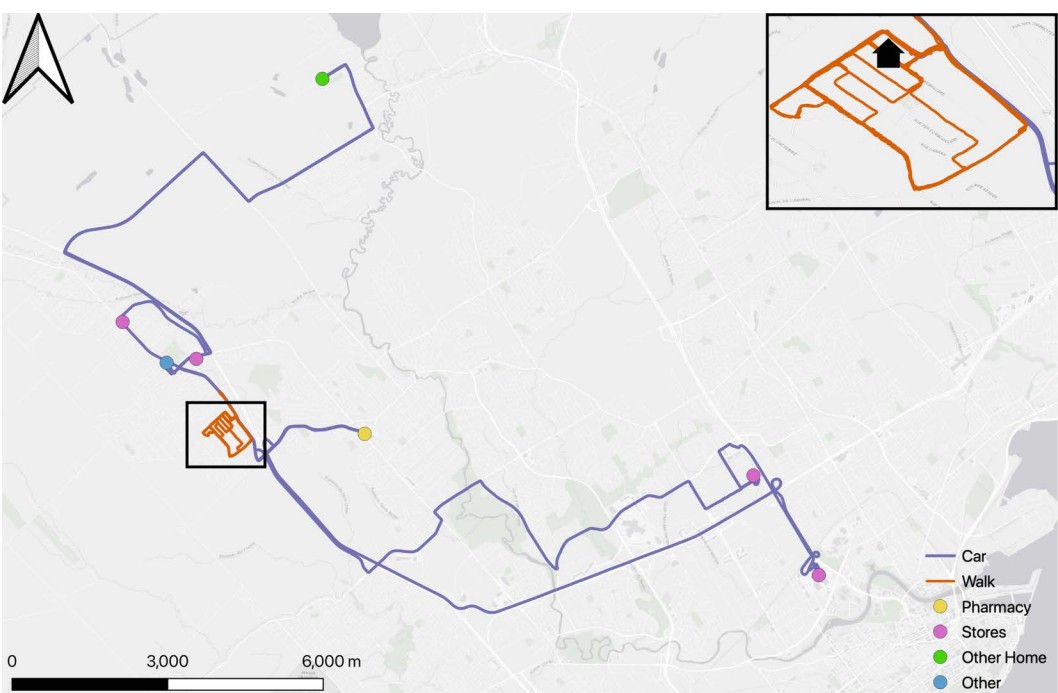

**Fig 2. Participant #1 Activity spaces map.**

**Adaptation strategies.** To cope with stress and navigate the obstacles, participants adopted regular routines with points of reference (n=8) as a guiding framework for their trips: "We get to walk down Heritage towards…and on Hammer Hill, towards Heritage Drive which is a city block away on the sidewalk and stop for a moment or two. And then we get to Heritage Drive and then walk back, that's about a 10-minute walk to 15 minutes" (P109).

**Covid-19.** Over the time the study was conducted, Alberta had declared a state of emergency due to the Covid-19 pandemic that lasted 21 months. The city of Calgary had periods of police-enforced curfew and lockdown in 2020–2021 [77]. Certain Alberta participants pointed out that the pandemic had lasting impact on mobility routines: "I used to do exercise over there, but I don't know. We don't go out to eat... Covid seems to… just make me shut down. I haven't done it" (P110).

## Activity space and mobility patterns

Of the 9 GPS maps generated, we selected 4 contrasting participants' maps—2 from each province (Participants 01 and 04 from Quebec, and Participants 106 and 111 from Alberta). The selection was based on their mobility patterns and modes of transportation, considering the frequency of outings from their houses and the distance traveled in kilometers. In Quebec, Participant 01 was less active, resulting in fewer activity spaces and a less elaborate mobility pattern in comparison with Participant 04, who had more varied and complex activity spaces and was more independent using all 3 means of transport, including the bus. In Alberta, Participant 106 was relatively inactive and used just one means of transportation while Participant 111 had multiple means of transportation and varied activity spaces. The GPS data generated allowed us to also create small inset maps to zoom in on activity spaces near the participants' residence.

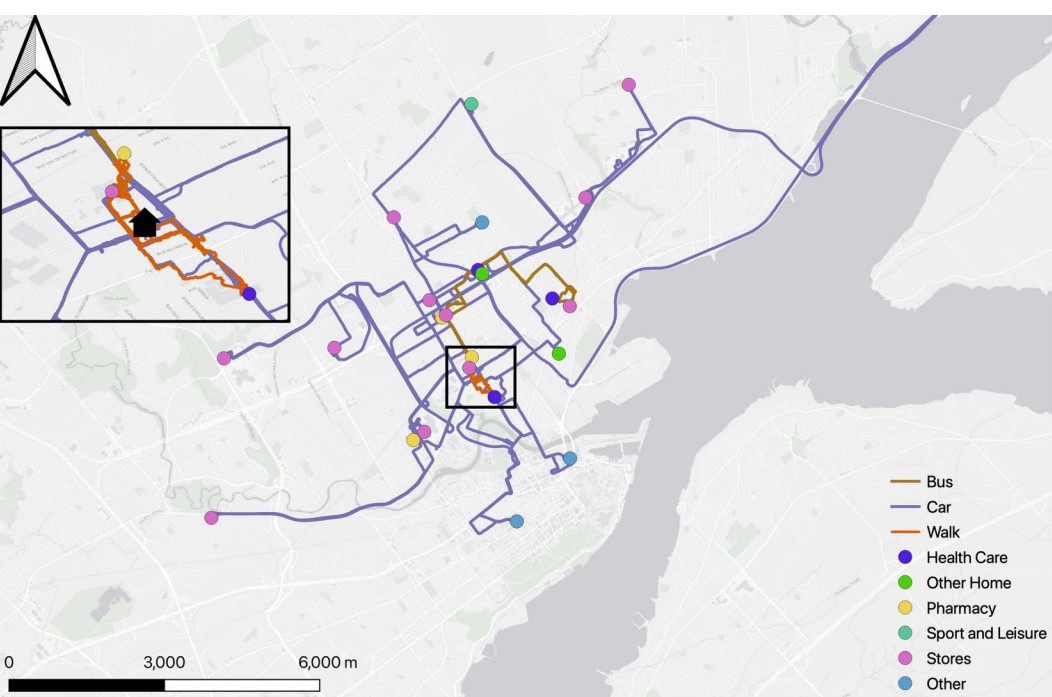

**Fig 3. Participant #4 Activity spaces map.**

**Activity spaces and mobility patterns maps in Quebec.** *Participant 1*: Based on the GPS analysis, Participant 1 incorporated walks into his daily routine. These are for the purpose of exercise rather than to reach a particular destination. The car journeys on the other hand, to the pharmacy and grocery stores, appeared to be for essential supplies or to visit friends or family. Fig 2 shows his activity spaces map.

Interviews confirmed these findings, but also provided a context for and a historical perspective on these car journeys: Participant 1 mentioned that he had lived in his neighborhood for the past 25 years and had witnessed its growth with the emergence of new buildings and streets. There were not many services available in his neighborhood and he had to drive longer distances from his house in search of services. Even though there were no services near his home, Participant 1 would only move if his needed to for health reasons: "I wouldn't exchange this neighborhood for another. Just change for change's sake, it's not worth it...... let's say you get sick, then you might move nearer to the hospital, then you might move to the city."

His daily journal showed that all trips made by car were with someone from his family. The GPS also suggested his walks lacked a predetermined and specific destination (the map showed no services on her route), and the interview indeed revealed that he preferred to walk around the neighborhood to see the mountains every day. His sense of attachment to his community, his enjoyment of the mountains, and his reluctance to relocate unless facing significant health issues points towards the complex interplay of personal, community and health considerations in the decision-making processes of older adults regarding their living arrangements.

*Participant 4*: Participant 4 had fewer and short walk than Participant 1, but according to GPS analysis her walks were to specific destinations: receive medical care or to go to the pharmacy. Fig 3 shows her activity spaces map.

She also took the bus and drives. She mentioned appreciating the proximity of services in their neighborhood, but also pointed out that one of the reasons she walked less is the lack of

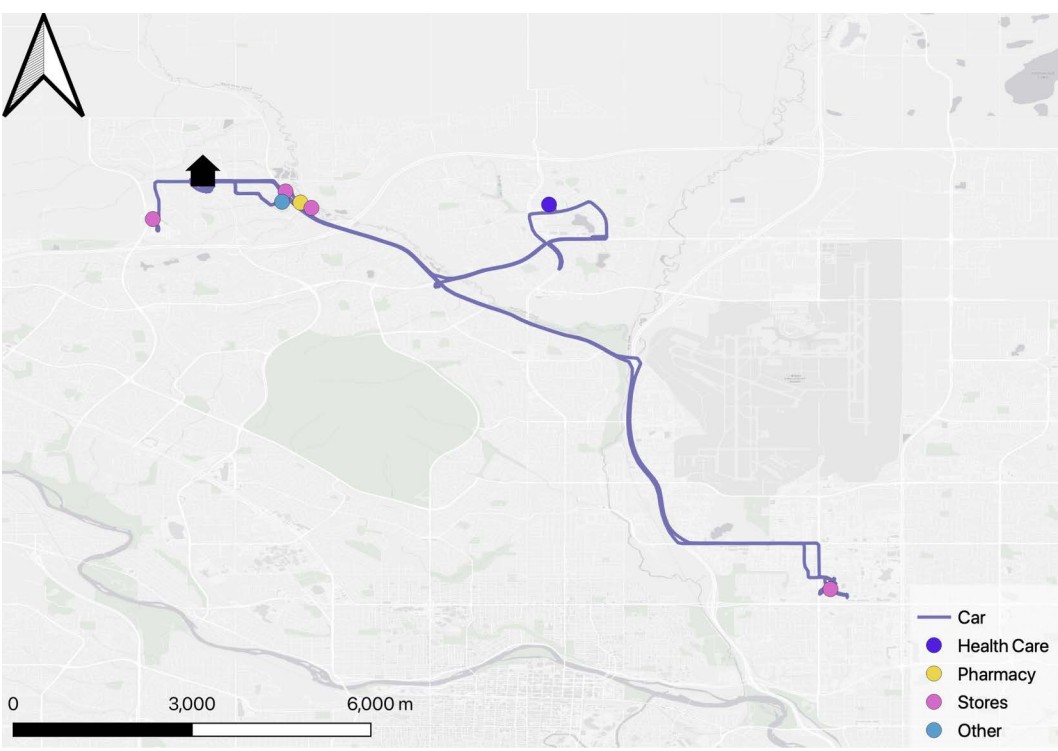

**Fig 4. Participant #106 Activity spaces map.**

places to rest during her walk. She was able to use the bus for certain closer destinations, but also made long car journeys. In contrast, there were more varied activities with Participant 4's than in Participant 1's. The journals of the Quebec participants revealed that Participant 1 undertook only planned activities during the 2 weeks, while half of Participant 4's activities were unplanned, suggesting that her easy access to neighborhood services and use of several means of transportation are more conducive to last minute outings. It could also reflect her personality or lifestyle.

Based on GPS data, Participant 4 often undertook extensive journeys by car. The daily journal showed that all these car trips were accompanied by another individual, whereas the walks and bus journeys were alone. This highlights that while more independent exploration is possible via walking and bus travel, driving may have required a companion. It was unknown, however whether Participant 4 was the driver or passenger Still, the unplanned activity spaces in this participants' life suggested an individual in robust health with aspirations to engage in a myriad of pursuits – a sentiment corroborated by the interview findings. Participant 4 indeed maintains an active circle of friends, affording her opportunities for unplanned and frequent visits and leisure in their company. Moreover, Participant 4's daily journal entries revealed a significant connection with her granddaughter and other children, demonstrating the importance of familial bonds in her life. The triangulation of data illuminated both her personal agency in pursuing her social interactions and her deeply rooted responsibilities within her family dynamic as motivations for dynamic aging in place. For Participant 4, these multiple physical and social assets and varied activity spaces and mobility patterns should be important elements in any future housing decisions.

**Activity spaces in Alberta.** *Participant 106*: In the case of the Alberta participants, there was a similar contrast between the 2 maps as between the two maps in Quebec. Participant

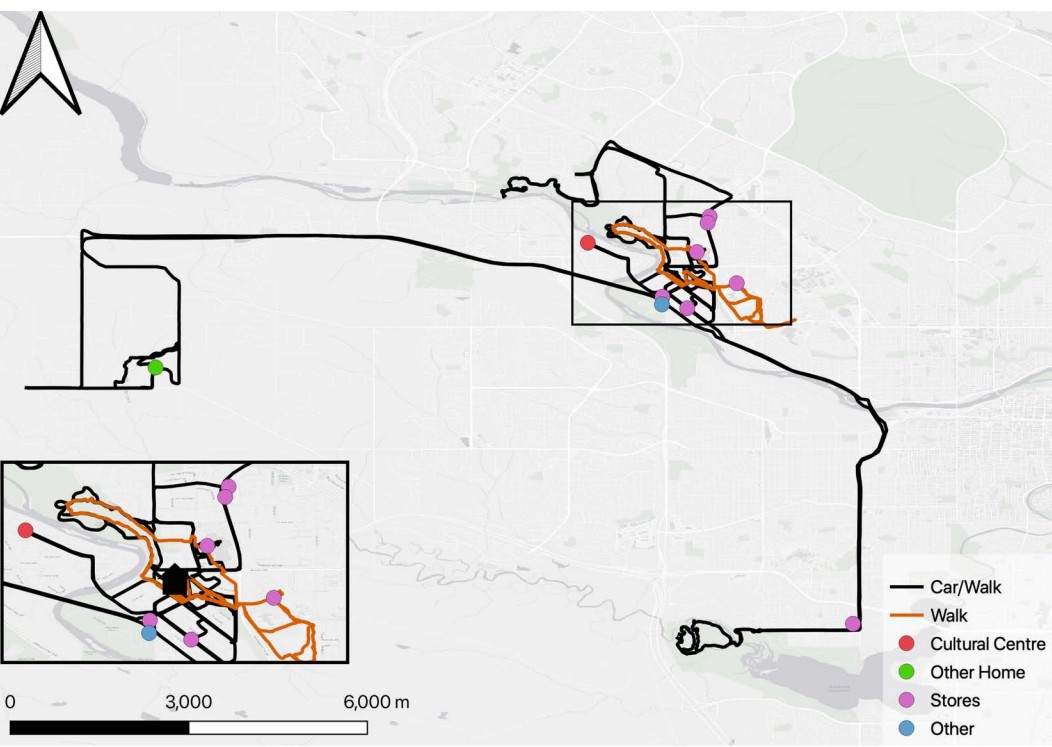

**Fig 5. Participant #111 Activity spaces map.**

106, over the course of 2 weeks, made a total of 5 car trips, all of which were essential, involving visits to medical care facilities, the pharmacy, and grocery stores. Additionally, there was a single stop at a church. Notably, this participant did not engage in any walking activities as presented in Fig 4.

Indeed, Participant 106 displayed locomotion difficulty from the start of the walking interview, indicating impediments to her mobility. She has great difficulty even leaving her own building: "Oh yes. I have to negotiate with my walker. These stairs are really hard to negotiate… And there's no railing."

From the interview we know that although Participant 106 is aware of the importance of exercising and desires to take walks, she does not engage in activities she would like to pursue because of the absence of physical aids that could assist with her mobility.

Visiting friends also becomes difficult, and her both her physical and social health suffer as a consequence, although she appreciates the offers of help from her neighbors: "It's amazing how many people, if they see you struggling, they just run over… It's very touching."

The effort required to overcome the physical barriers to mobility experienced by Participant 106 not only restricts her access to social interactions but also exacerbates the sense of social isolation that can accompany aging. This participant had made the decision to move into an adult-only building, and while there may be some advantages to this housing option, perhaps the full extent of the social and physical assets of her previous residence and the potential assets in the new residence had not been fully taken into account in this housing decision.

*Participant 111*: Participant 111, in contrast, frequently drives to different places. From the GPS perspective, Participant 111 provided more data and more information, presenting a panorama of her activity spaces and robust social health as presented in Fig 5.

At first sight, Participant 111's mobility patterns seemed less strategic than those of Participant 106. Because there appeared to be less routine, data building was especially important in the case of Participant 111. The interviews revealed that her dog is an important part of her life: "My life is all about dog walks. I didn't really have to go anywhere." These walks give her a sense of meaning, increase her mobility and motivate her to walk extensively in her neighborhood (orange line in Fig 5). Participant 111 also leads an engaging and spontaneous social life, frequently visiting friends' homes and organizing walks in different parks, both individually and with a group. As shown on the map, Participant 111 uses a mixed mode of transportation, using a car to travel to the park for walks. The interviews revealed that she picks up her friends on the way. This active social involvement contributes to the extensive mobility displayed in the GPS data. Overall, Participant 111's sociability and mobility patterns highlight an active lifestyle, positioning her as a central figure within her social circle. As with Participant 4, these multiple physical and social assets, varied activity spaces and creative mobility patterns should be important elements in any future housing decisions.

The convergence of sources provided new insights, revealing that the use of complementary sources sheds light not only on variations in their modes of transportation, and how the older adults navigated these spaces, but also the social aspects of their lives not captured by the individual data sources.

## Discussion

We used the concept of activity space to explore the mobility patterns and social health of Canadian older adults living at home, which could inform decisions about housing options which, in turn, could be considered in the design of decision aids for making housing decisions. Large variations were seen with mobility patterns in distances travelled over the observation period, with Quebec participants travelling greater distances. Cars were the most frequently reported means of transportation with the primary destination being going to a store. Also, most respondents travelled alone. We found that the most frequently reported social asset was the proximity of family members and neighbors and attachment to neighborhood, while the most frequently reported physical asset were benches and neighborhood institutions, as also found in the sister study in the Netherlands [44]. The most frequently reported social and physical obstacles were related to winter weather and mobility. Participants mentioned points of reference and physical supports such as canes as adaptation strategies to address mobility obstacles. Lastly, the merging of sources provided new insights, revealing variations in participants' modes of transportation and how they navigated these spaces, and also highlighting social aspects not captured by the individual data sources.

These results lead us to make the following observations.

First, activity space maps illustrated a marked difference in number and type of activities between Alberta and Quebec which may speak to both the built-environment, green-space and the physical infrastructure. In both provinces, stores were the most frequently visited places. Our results show resonance with other studies [78,79] in which the primary focus of trips outside the home for older adults is going to stores. However, in Quebec, participants were more inclined to visit family and friends and participate in cultural events, also found in another Canadian study on participation in social activities [80]. In Quebec, independence and autonomy were reinforced by the proximity of walkable neighborhoods with more available and accessible services for older adults. As our results also demonstrated that most trips outside the home were made alone, walkable neighborhoods contributed to enabling them to maintain their autonomy. Even though the car was the predominant mode of transportation in both provinces, in Quebec, participants routinely used the car in combination with another

means of transportation, such as walking or taking the bus, whereas in Alberta, participants mostly relied on the car for their activities. This may suggest a lack of public transportation infrastructure or conversely a stronger car culture in Alberta, a province associated with the oil industry [81]. It may also suggest stronger presence of family in Quebec, a province that historically has much higher intraprovincial retention than Alberta. Quebec's attachment to the French language also means they are more likely to remain within the province [82]. Furthermore, participants with more varied activity spaces were those who utilized more than one means of transportation. This implies that a broader range of transportation options for older adults which provides greater access to more places. According to the World Health Organization (WHO) [83], mobility is fundamental for older adults, and preserving it stands as the most effective strategy for remaining actively engaged within their community. Designers of decisions aids for housing decisions should take mobility into account as well as comparison of existing and future transport options, along with importance of car transportation, when they present older adults with housing options.

Second, family constituted a significant social asset, exerting a substantial influence on their social networks and social support [84]. Other studies confirm that the absence of familial support negatively affects the social connections and overall social well-being of older adults and their life satisfaction [85]. In terms of autonomy in daily life, even if the older adult does not need direct aid from a family member, their offer of availability improves well-being and contributes to their perception of emotional support [86]. As hinted in our results, this availability can also sometimes be invasive. However, in general, the proximity and accessibility of family are important considerations in making housing decisions. The interviews also revealed a strong feeling of community attachment to the neighborhood and neighbors, with many participants echoing similar narratives of attachment, underscoring the importance of a social fabric that has developed over time. Participants' emotions about their neighborhoods are tightly interwoven with the decision-making process about housing. The concept of neighborhood goes beyond being a mere geographical contingency; it reveals significance to a community and expresses its local culture and has the potential to shape collective identity and increase social interactions. In the field of gerontology research, the neighborhood category is progressively gaining recognition as a primary and decisional factor in the health and well-being of older adults [87–89]. In our research, we observed that residents in Alberta needed to travel greater distances to reach stores for their shopping needs. This stands in contrast to participants from Quebec, who underscored the importance of local establishments such as grocery stores. They emphasized that the ability to walk to nearby stores (i.e., walkability) is a determining factor for the well-being of older adults. Regarding physical assets, our findings suggest that older adults adapt their walking routes based on the presence of benches, emphasizing the crucial role of a surrounding environment that takes their mobility needs into account. Ottoni et al. [90], who conducted research in Vancouver, Canada, demonstrated that benches positively impact the mobility of older adults. Their contribution extends beyond being a mere physical asset, they change behaviors. This was also demonstrated by Sturge et al. [44], who noted that while they are necessary breaks for older walkers, the accidental encounters they provide create additional intergenerational social integration. Benches enhance the utilization and enjoyment of green areas, also elevating the social capital of older adults [90]. As the built environment plays a significant role in influencing the mobility of older adults [44,79,91–94], their housing decisions should consider factors such as the accessibility of benches in the immediate surroundings.

An accessible neighborhood offers both a physical and a social asset as well, in terms of its institutions (e.g., libraries, grocery stores), which play a crucial role as they provide important venues for social engagement. For example, other studies [95,96] have indicated that public

libraries, in particular, foster social connections among older adults. In addition, as some participants noted, their visits were not solely to interact with known individuals but also to engage with the broader community and new people, emphasizing the importance of access to neighborhood institutions for fulfilling the inherent human need for social contact and the formation of new connections.

Third, obstacles to social health included the weather which was frequently expressed as an impediment to their activities outside the home. Canadian winter weather decreases social connection and increases the risk of social isolation [97]. The reduction of the walkability of roads by snow and ice directly influences older adult's physical health and well-being [98]. Many have adapted their activities to overcome the challenge by adjusting their routes or using canes and crampons, which can increase the risk of falls. Preventing falls is one of the most significant concerns in terms of public health for older adults. According to the World Health Organization (WHO), one-third of older adults fall each year. In older adults aged 70 or older, this percentage varies between 32 and 70 [99,100]. Mondor et al. [101] demonstrate that in winter, especially during freezing rain and snowstorms, there is an increase in fall-related injuries in older adults, posing a serious public health issue in Canada. In both Alberta and Quebec, freezes and thaws can occur in quick succession, covering the ground in a sheet of ice. This is evident as the number of hospital admissions for fall-related issues is growing among those aged 65 and older [102]. Walkability for older adults decreases in winter, and decision aids about housing decisions for older adults in places with severe winters such as Quebec and Alberta (where temperatures of -35 degrees C are not uncommon) need to consider the impact of winter conditions and the adequacy of local snow-clearing on their housing options, as well as whether their existing coping strategies are transferable.

Fourth, by combining and building on different data sources, we were able to capture various intersecting layers in the lives of our participants, providing a better understanding of their needs for information regarding upcoming housing decisions or for the design of decision aids. Considering only GPS data as the primary and objective data provided information of participants' routines in terms of places, distances, and means of transportation, but cross-referencing these data with the interview data enabled us to explore nuances not captured initially. For example, Participant 106 revealed during her interview that she was aware of the importance of walking and mobility; however, her building, exclusively for older adults, restricted mobility due to COVID. The information regarding restricted mobility in her building would have been an important consideration prior to moving there, and such questions should be included in decision aids. The combined data of Participant 111 also demonstrated that with GPS data alone, it would be difficult to understand her mobility patterns: in fact, many of her car trips involved taking friends with her to go to their walking group and walking her dog. These activities suggest a whole additional range of mobility-related social and physical assets of the older person in their current location that should be taken into account in any future design of decision aids about housing, as a move might affect these more subtle aspects of their quality of life. Otherwise, these precious assets, so easily taken for granted, may only be appreciated in retrospect. The best people to inform decision aid designers of these more subtle aspects of their lives are the older people themselves. Designers cannot create adequate decision aids for older adults without involving them iteratively throughout the design process [103]. One contributing factor may be the perception of older adults as passive users [104,105] in the design process of tools aimed at them. By altering their role and positioning this population as protagonists in the conceptualization phase [106], the development process should enable them to explain their needs and difficulties and express what they currently appreciate most about their lives [107]. This will ultimately empower them as co-creators of the tools that help them make their difficult housing decisions.

Lastly, using multiple data collection methods and building one on top of another [54] offered a unique perspective on older adults' space and mobility patterns. Early mobility studies primarily focused on either GPS data or self-reported data, without integrating these approaches in the same analysis [53]. However, with progress in technology and new GPS processing methods, recent studies have begun to integrate both qualitative methods (such as travel diaries, walking interviews) and quantitative GPS data, demonstrating the applicability and coherence of adopting this method [70,108–111]. Although this convergent mixed-methods study [53] diverges from the strictest definition of a mixed-methods study, it gave added nuance to our findings. The social health, for example, cannot be measured solely through GPS. It requires contact and trust between the research team and participants, developed through two interview sessions that were able to reveal nuances not discernible through GPS alone. The small sample gave us the richness of information but limited generalizability. However, we did not seek the big picture, we explored the culture, geography and language of the two very different provinces to explore the heterogeneity of older adults' mobility and social assets, suggesting that the future design of decision aids for housing decisions must be flexible to such differences.

## Limitations of the study

Our study has some limitations. Not all participants were able to use the GPS properly or fill out the daily journal, which limited the analysis. During the interview, some participants had negative impressions of wearing the GPS devices. Some found them uncomfortable, inconvenient, awkward ("It was amazing that my equipment did not land in the toilet"), and that the device attracted unwanted attention. One participant had privacy concerns ("it's just too personal"), and another felt they were being tested by researchers comparing the daily journal with the GPS data. Future research assessing mobility patterns will benefit from less intrusive technology (e.g., smart watches) and improved methods for encouraging participants to complete data entries in the daily journal. In addition, the Covid-19 pandemic hampered data collection in Alberta. However, after the isolation period, we resumed our research and asked participants about their mobility patterns and experiences during the pandemic, shedding light on another perspective not initially planned in this study.

Second, our sample sizes were small, which limits the extent to which our findings can be generalized. However, our data merging followed a qualitatively driven approach, and sample sizes in qualitative research are typically smaller than in quantitative research [110,111]. Despite the smaller sample, our sampling strategy allowed us to work with "information-rich" participants [112]. Importantly, our goal was not to generalize but to explore a diverse range of experiences related to social health and mobility patterns from the direct experiences of older adults themselves.

## Conclusions

This mixed-methods exploration of the activity spaces and mobility patterns of older adults in Quebec and Alberta revealed that despite a decline in autonomy, numerous existing social and physical assets associated with mobility make important contributions to their social health, such as access to several means of transportation and to public institutions, familiarity with their neighborhoods, and the proximity of friends and family.

This study underlines the importance of integrating mobility patterns and their associated social assets into housing decision frameworks. It provides guidance for the development of user-informed decision support tools, tailored specifically to Canadian older adults' housing decisions, that consider older adults' existing physical and social assets in common real-life

situations. These actionable insights can inform the creation of tools that promote healthier aging and facilitate more person-centered housing decisions.

## Supporting information

**S1 Appendix. Code frequencies.**
(DOCX)

## Acknowledgments

We thank Louisa Blair for her manuscript revision, Souleymane Gadio and Georgina Dofara for their support in the descriptive analysis. We appreciate the support of Greybox Solutions in providing us with an in-kind contribution.

## Author contributions

**Conceptualization:** Diogo Mochcovitch, Allyson Jones, Joshua Goutte, Karine V. Plourde, Louise Meijering, Jodi Sturge, Stéphane Roche, Sabrina Guay-Bélanger, France Légaré.

**Data curation:** Diogo Mochcovitch, Allyson Jones, Joshua Goutte, Karine V. Plourde, Roberta de Carvalho Corôa, France Légaré.

**Formal analysis:** Diogo Mochcovitch, Allyson Jones, Joshua Goutte, Karine V. Plourde, Roberta de Carvalho Corôa, Marie Elf, Louise Meijering, France Légaré.

**Funding acquisition:** Allyson Jones, Marie Elf, Louise Meijering, Pierre Bérubé, France Légaré.

**Investigation:** Diogo Mochcovitch, Allyson Jones, Joshua Goutte, Karine V. Plourde, Roberta de Carvalho Corôa, France Légaré.

**Methodology:** Diogo Mochcovitch, Karine V. Plourde, Louise Meijering, Jodi Sturge, France Légaré.

**Project administration:** Allyson Jones, France Légaré.

**Resources:** Allyson Jones, Pierre Bérubé.

**Supervision:** Diogo Mochcovitch, Karine V. Plourde, Marie Elf, Sabrina Guay-Bélanger, France Légaré.

**Validation:** Diogo Mochcovitch, Allyson Jones, Roberta de Carvalho Corôa, Marie Elf, Louise Meijering, Jodi Sturge, Stéphane Roche, Sabrina Guay-Bélanger, France Légaré.

**Visualization:** Diogo Mochcovitch, Allyson Jones, Joshua Goutte, Karine V. Plourde, France Légaré.

**Writing – original draft:** Diogo Mochcovitch, Joshua Goutte, Karine V. Plourde, France Légaré.

**Writing – review & editing:** Diogo Mochcovitch, Allyson Jones, Joshua Goutte, Karine V. Plourde, Roberta de Carvalho Corôa, Marie Elf, Louise Meijering, Jodi Sturge, Pierre Bérubé, Stéphane Roche, Sabrina Guay-Bélanger, France Légaré.

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
