## [Decision Letter · Decision Letter 0]

10 Oct 2024

PONE-D-24-37265Exploring mobility patterns and social health of older Canadians living at home to inform decision aids about housing: a mixed-methods studyPLOS ONE

Dear Dr. Légaré,

Thank you for submitting your manuscript to PLOS ONE. After careful consideration, we feel that it has merit but does not fully meet PLOS ONE’s publication criteria as it currently stands. Therefore, we invite you to submit a revised version of the manuscript that addresses the points raised during the review process.

We look forward to receiving your revised manuscript.

Kind regards,

Li-Pang Chen

Academic Editor

PLOS ONE

2. We note that you have indicated that there are restrictions to data sharing for this study. For studies involving human research participant data or other sensitive data, we encourage authors to share de-identified or anonymized data. However, when data cannot be publicly shared for ethical reasons, we allow authors to make their data sets available upon request. For information on unacceptable data access restrictions, please see http://journals.plos.org/plosone/s/data-availability#loc-unacceptable-data-access-restrictions. Before we proceed with your manuscript, please address the following prompts: a) If there are ethical or legal restrictions on sharing a de-identified data set, please explain them in detail (e.g., data contain potentially identifying or sensitive patient information, data are owned by a third-party organization, etc.) and who has imposed them (e.g., a Research Ethics Committee or Institutional Review Board, etc.). Please also provide contact information for a data access committee, ethics committee, or other institutional body to which data requests may be sent. b) If there are no restrictions, please upload the minimal anonymized data set necessary to replicate your study findings to a stable, public repository and provide us with the relevant URLs, DOIs, or accession numbers. Please see http://www.bmj.com/content/340/bmj.c181.long for guidelines on how to de-identify and prepare clinical data for publication. For a list of recommended repositories, please see https://journals.plos.org/plosone/s/recommended-repositories. You also have the option of uploading the data as Supporting Information files, but we would recommend depositing data directly to a data repository if possible. Please update your Data Availability statement in the submission form accordingly.

4. Please ensure that you refer to Figure 2, 3, 4, and 5 in your text as, if accepted, production will need this reference to link the reader to the figure.

5. Please include a caption for figure 6.

Additional Editor Comments:

Dear Authors:

The referee has reviewed your manuscript and has provided the review report. The feedback from the referee is positive and we would like invite you to resubmit the revision. Please carefully follow the comments raised by the referee and then revise the manuscript.

Reviewers' comments:

Reviewer's Responses to Questions

**Comments to the Author**

1. Is the manuscript technically sound, and do the data support the conclusions?

Reviewer #1: Partly

2. Has the statistical analysis been performed appropriately and rigorously? 

Reviewer #1: N/A

3. Have the authors made all data underlying the findings in their manuscript fully available?

Reviewer #1: Yes

4. Is the manuscript presented in an intelligible fashion and written in standard English?

Reviewer #1: No

5. Review Comments to the Author

Reviewer #1: Comments to the Author

This research has the potential to assist older Canadians in making housing decisions; however, I would like to clarify the following points:

・The introduction part:

1: “Mobility and housing” seem to be the core terms of your study. However, the scope of what you mean by those two terms needs to be clarified. In addition, the hypothesis from Page 5, Line 126 is not specific, as it only states the expectation that “novel insights” will be obtained. Therefore, the overall goal of the study is not clear.

Please elaborate on the definition and scope of the terms “mobility,” “housing,” or “housing-related decisions” and what value they bring.

2: There is a lack of articles; thus, some parts are unclear what they indicate.

In particular, the word “mobility” is not clear who and what kind of mobility. For example, on Page 5, Line 125: Based on these gerontological concepts, we sought a new perspective on how mobility patterns and activity spaces affect the social health of older adults.

3: Some of the citations need to be corrected. Especially in page 5, Lines 120-122, I have never seen a citation attached to your question. Please stick to academic writing.

4: Page4, Line 99-100

　I don't know what housing transitions indicate. Even if this is some element related to housing, there is not enough evidence to strongly connect it to social health with “must therefore” in the previous sentences.

4: Page 5, Line 132

　Instead of “suggest” at the introduction stage, try to state some findings based on the method, result, and discussion.

5: Page5, Line 136-138

Here you seem to suggest something important as the end of Introduction, but I don't know what you are stating.

6: In the Sociodemographic and other self-reported information part, please describe exactly how and what variable information you obtained. For example, there is a classification of health as poor, fair, etc., but it is not clear how it was divided.

7: In the data analysis part, please briefly explain what kind of analysis method you used to look at the relationship between what and what. Perhaps what you want to show is that mobility patterns are associated with social health, which you mention in the conclusion. Please explain here the analytical method used to provide the evidence for this.

The current description only states the purpose of combining quantitative and qualitative data and the method of data collection. It also does not state what kind of target is the subject of the analysis. Therefore, it is not possible to determine whether the results cover the entire n=20 or just a portion of it.

Result part

8: It is not clear which population (n=20, n=14, or n=7) was analyzed throughout. For example, for which population are the results for Page 15, Line326 migration?

9: Where does the discussion part start? What is the point you want to make in the discussion part? Please briefly summarize how it relates to key words such as housing decision.

Conclusion part

10: Even if mobility patterns are associated with social health, are they related to designing decisions or housing transitions? I do not understand the causal relationship.

6. PLOS authors have the option to publish the peer review history of their article (what does this mean? ). If published, this will include your full peer review and any attached files.

**Do you want your identity to be public for this peer review?** For information about this choice, including consent withdrawal, please see our Privacy Policy .

Reviewer #1: **Yes: ** Takeshi Endo

---

## [Author Response · Author response to Decision Letter 1]

14 Nov 2024

Editor’s comments

1. Please ensure that your manuscript meets PLOS ONE's style requirements, including those for file naming. The PLOS ONE style templates can be found at Style template1 and Style template2

Response: Thank you for the comment. We noticed that in some places we used italics, and we have replaced them with the Style template as recommended.

2. We note that you have indicated that there are restrictions to data sharing for this study. For studies involving human research participant data or other sensitive data, we encourage authors to share de-identified or anonymized data. However, when data cannot be publicly shared for ethical reasons, we allow authors to make their data sets available upon request. For information on unacceptable data access restrictions, please see http://journals.plos.org/plosone/s/data-availability#loc-unacceptable-data-access-restrictions. Before we proceed with your manuscript, please address the following prompts: a) If there are ethical or legal restrictions on sharing a de-identified data set, please explain them in detail (e.g., data contain potentially identifying or sensitive patient information, data are owned by a third-party organization, etc.) and who has imposed them (e.g., a Research Ethics Committee or Institutional Review Board, etc.). Please also provide contact information for a data access committee, ethics committee, or other institutional body to which data requests may be sent. b) If there are no restrictions, please upload the minimal anonymized data set necessary to replicate your study findings to a stable, public repository and provide us with the relevant URLs, DOIs, or accession numbers. Please see http://www.bmj.com/content/340/bmj.c181.long for guidelines on how to de-identify and prepare clinical data for publication. For a list of recommended repositories, please see https://journals.plos.org/plosone/s/recommended-repositories. You also have the option of uploading the data as Supporting Information files, but we would recommend depositing data directly to a data repository if possible. Please update your Data Availability statement in the submission form accordingly.

Response: Thank you for the opportunity to clarify our data availability statement. We confirm that there are restrictions on publicly sharing part of the raw data set, even in de-identified form, as the GPS data contains potentially identifying information that could be used to retrace participants' locations. These restrictions were imposed by the Research Ethics Committee (Comité d'éthique de la recherche du CIUSSS de la Capitale-Nationale) to protect participant confidentiality and ensure compliance with institutional policies regarding sensitive human participant data.

However, a substantial portion of the raw data set is available in anonymized form in the supporting files, as indicated in the main text.

We revised our previous statement to clarify that future researchers should contact the Ethics Committee directly. The updated statement on the PLOS ONE platform is as follows:

“As data were collected and stored, they were all de-identified using anonymous identifiers to ensure confidentiality. However, due to the nature of GPS data, there remains a risk of potential re-identification within raw datasets. The Ethics Committee approved the collection and analysis of the data only for the specific study and not for other purposes. The Ethics Committee requires that the data collected remains securely stored and not be shared.

Researchers interested in accessing this data may submit a request to the Comité d'éthique de la recherche du CIUSSS de la Capitale Nationale at Simon Trempe at 555 boul. Wilfrid-Hamel, bureau E-115, Québec (Québec), G1M 3X7, or via email at simon.trempe.ciussscn@ssss.gouv.qc.ca (more information on the ethics committee is available at https://www.ciusss-capitalenationale.gouv.qc.ca/mission-universitaire/recherche/ethique-recherche/sante-population-premiere-ligne). We report all data relevant to answer our research question in the paper.”

Response: We added the full Ethics statement as indicated included on the PLOS ONE platform. We created a new section after 'Study Design and Context' on page 7, lines 168-172, and included the following paragraph:

“Ethical approvals were obtained from the Health Ethics Research Board at the University of Alberta (Pro00087478) for Alberta and from the Integrated University Health and Social Services Centres (CIUSSS) de la Capitale Nationale (2019-1519) for Quebec. Written and informed consent was obtained and secured from all participants prior to their enrollment in the study.”

4. Please ensure that you refer to Figure 2, 3, 4, and 5 in your text as, if accepted, production will need this reference to link the reader to the figure.

Response: Thanks for the comment. We reviewed and added the reference in the text for Figures 2, 3, 4 and 5.

5. Please include a caption for figure 6.

Response: The title “Figure 6” was a mistake, and we have replaced it by “Figure 5”.

Additional Editor Comments:

Dear Authors:

The referee has reviewed your manuscript and has provided the review report. The feedback from the referee is positive and we would like invite you to resubmit the revision. Please carefully follow the comments raised by the referee and then revise the manuscript.

Response: Thank you for your valuable comment; we greatly appreciate it. We hope that our revisions meet the necessary standards for publication.

Reviewers’ comments

Reviewer #1: Comments to the Author

This research has the potential to assist older Canadians in making housing decisions; however, I would like to clarify the following points:

・The introduction part:

1: “Mobility and housing” seem to be the core terms of your study. However, the scope of what you mean by those two terms needs to be clarified. In addition, the hypothesis from Page 5, Line 126 is not specific, as it only states the expectation that “novel insights” will be obtained. Therefore, the overall goal of the study is not clear. Please elaborate on the definition and scope of the terms “mobility,” “housing,” or “housing-related decisions” and what value they bring.

Response: Thank you for your valuable and insightful comment.

We revised the paragraph where 'housing' is first mentioned and added a definition on page 3, lines 77-80.

“In our previous work, we found that housing decisions among older adults, i.e., the decision-making process of choosing whether they will age at home or move into institutional residential care [5, 6], are the most prevalent among the difficult decisions that they face [5, 7, 8].”

For this paragraph, we have added the following references:

6. Roy N, Dube R, Despres C, Freitas A, Legare F. Choosing between staying at home or moving: A systematic review of factors influencing housing decisions among frail older adults. PLoS One. 2018;13(1):e0189266.

8. Claudia Lai PH, Karine V. Plourde, Lily Yeung, France Légaré. Home care providers' perceptions of shared decision-making with older clients (and their caregivers): A cross-sectional study. Nursing & health sciences. 2021;24:487-98.

Additionally, we added a second paragraph on page 3, lines 81-86, to contextualize why it is important to examine housing decisions and the experiences of older adults.

“Although there are many conceptual frameworks regarding housing decisions [9-13], social and emotional concerns of older adults, as well as their existing physical and social assets, are often overlooked [6, 14]. As a result, a housing decision may not fully align with their preferences, increasing their distress [5]. For example, an older adult with a habit of morning walks may relocate to an area where it is difficult to walk due to the proximity of busy roads.”

For this excerpt, we have added the following references:

9. ES L. A theory of migration. Demography. 1966;3:47–57.

10. Lawton MP NL. Ecology and the aging process. VI ed. Washington: American Psychological Association. 619–74 p.

11. Litwak E, Longino CF, Jr. Migration patterns among the elderly: a developmental perspective. Gerontologist. 1987;27(3):266-72.

12. RF W. Why older people move: Theoretical issues. Research on Aging. 1980;2(2):141-54.

13. RM A. A behavioral model of families' use of health services. Studies CfHA, editor. Chicago: University of Chicago; 1968.

Regarding the term “mobility,” we have added a definition previously used by a team member in another article (Meijering, 2021). This modification can be found on page 5, lines 108-111.

“Mobility plays a key role in social health. While some definitions of mobility focus solely on physical ability, such as moving oneself from one place to another [24, 25], in this study, we consider that mobility also encompasses the purpose behind the movement, such as social interactions, family engagements, work, and leisure activities [26].”

The reference we have added:

26. Meijering L. Towards meaningful mobility: a research agenda for movement within and between places in later life. Ageing & Society. 2021;41(4):711-23.

Regarding the hypothesis and "novel insights," we appreciate your comment and have rewritten the overall goal of the study to make our point more directly. The updated sentence can be found on page 6, lines 137-143.

“Following an earlier exploratory study [22, 41, 42], we hypothesized that mobility and social health are importance dimensions to consider in housing decisions. Therefore, with the overall goal of informing the design of decision aids, we aimed to map a range of activity spaces of older adults in two provinces of Canada. By using GPS data combined with subjective narrative sources (journals and interviews), to highlight the importance of considering older adults’ existing social and physical assets when making housing decisions.”

2: There is a lack of articles; thus, some parts are unclear what they indicate. In particular, the word “mobility” is not clear who and what kind of mobility. For example, on Page 5, Line 125: Based on these gerontological concepts, we sought a new perspective on how mobility patterns and activity spaces affect the social health of older adults.

Response: We presume you mean articles in the grammatical sense. We have revised the paper and added articles where they are missing. Additionally, we have added more complete definitions of terms as outlined above.

3: Some of the citations need to be corrected. Especially in page 5, Lines 120-122, I have never seen a citation attached to your question. Please stick to academic writing.

Response: We have revised this paragraph so that it is no longer phrased as a series of questions. The updated version is located on pages 5-6, lines 128-134.

“For instance, it is important to identify the social supports, such as friends and family, that currently help maintain an older adult’s social health [39]. Or volunteering in their community may provide them with a healthy sense of self-worth, and their mobility may make this possible [40]. Therefore, it is important to understand the motivations that help them remain mobile and maintain their physical health and, if they decide to move elsewhere, to ask whether these social and physical assets will still be available to them.”

4: Page4, Line 99-100

　I don't know what housing transitions indicate. Even if this is some element related to housing, there is not enough evidence to strongly connect it to social health with “must therefore” in the previous sentences.

Response: Thank you for the comment. We have removed the reference to “housing transitions” to avoid confusion and now refer only to housing decisions, as defined above.

4: Page 5, Line 132

　Instead of “suggest” at the introduction stage, try to state some findings based on the method, result, and discussion.

Response: Thank you for the suggestion. As this sentence is related to your first question about the hypothesis, we addressed them together, revising the language to ensure the statement is more direct and concise, making it clearer and easier for readers to understand the main point. We believe these adjustments enhance the overall readability and effectiveness of the content.

5: Page5, Line 136-138

Here you seem to suggest something important as the end of Introduction, but I don't know what you are stating.

Response: As mentioned above, we have clarified the hypothesis and aims of the paper in the last paragraph of the Introduction.

6: In the Sociodemographic and other self-reported information part, please describe exactly how and what variable information you obtained. For example, there is a classification of health as poor, fair, etc., but it is not clear how it was divided.

Response: Thanks, we appreciate this comment. We added in the Sociodemographic and other self-reported information section (page 8-9, lines 194-202) the following sentence:

“For self-reported health status, we used a 5-point Likert scale [51], allowing participants to rate their health based on subjective experience, with options ranging from poor, fair, good, very good, to excellent. For questions regarding assistance required with daily tasks, we based our items on the Instrumental Activities of Daily Living (IADL) Scale, which assesses the ability to perform complex everyday tasks (e.g., managing finances and medications) evaluating whether the participants receive or provide help with those tasks [52]. Regarding technology use, we asked participants to specify the types of technology they typically use in their daily life.”

We consulted the following references:

51. Sullivan GM, Artino AR, Jr. Analyzing and interpreting data from likert-type scales. J Grad Med Educ. 2013;5(4):541-2.

52. Gold DA. An examination of instrumental activities of daily living assessment in older adults and mild cognitive impairment. J Clin Exp Neuropsychol. 2012;34(1):11-34.

7: In the data analysis part, please briefly explain what kind of analysis method you used to look at the relationship between what and what. Perhaps what you want to show is that mobility patterns are associated with social health, which you mention in the conclusion. Please explain here the analytical method used to provide the evidence for this.

The current description only states the purpose of combining quantitative and qualitative data and the method of data collection.

It also does not state what kind of target is the subject of the analysis. Therefore, it is not possible to determine whether the results cover the entire n=20 or just a portion of it.

Response: Thanks for these questions.

Following the initial statement of our data analysis goals, we have included an explanation of our approach to analyzing the data, located on pages 10-11, lines 236-250.

“We conducted this analysis in a hierarchical manner, layering one source upon another, with GPS data and interviews serving as primary sources, complemented by daily journal entries. We chose this approach due to the building nature of the data [45], where the interviews deepened our understanding of the previous experiences captured by the other data sources. For instance, GPS data may reveal a participant’s choice to take a longer route to the supermarket instead of a shorter one, while interview data could indicate that this choice serves as a coping strategy to avoid a noisy avenue that causes stress. Given that we adopted a definition of mobility that considers not only the physical aspects of travel but also other factors, this iterative feedback process complemented and integrated the data sources. The combination of methods allowed us to produce a comprehensive analysis of social health in the mobility patterns of older adults [44]. For participants who did not provide all primary data sources (e.g., those who did not us

---

## [Decision Letter · Decision Letter 1]

8 Dec 2024

PONE-D-24-37265R1Exploring mobility patterns and social health of older Canadians living at home to inform decision aids about housing: a mixed-methods studyPLOS ONE

Dear Dr. Légaré,

Thank you for submitting your manuscript to PLOS ONE. After careful consideration, we feel that it has merit but does not fully meet PLOS ONE’s publication criteria as it currently stands. Therefore, we invite you to submit a revised version of the manuscript that addresses the points raised during the review process.

We look forward to receiving your revised manuscript.

Kind regards,

Li-Pang Chen

Academic Editor

PLOS ONE

Additional Editor Comments:

The referee reviewed your revision and additionally raised some comments to the manuscript and recommended "Major Revision" again. I suggest the authors to follow the referee's comment and revise this manuscript carefully.

Reviewers' comments:

Reviewer's Responses to Questions

**Comments to the Author**

1. If the authors have adequately addressed your comments raised in a previous round of review and you feel that this manuscript is now acceptable for publication, you may indicate that here to bypass the “Comments to the Author” section, enter your conflict of interest statement in the “Confidential to Editor” section, and submit your "Accept" recommendation.

Reviewer #1: All comments have been addressed

2. Is the manuscript technically sound, and do the data support the conclusions?

Reviewer #1: No

3. Has the statistical analysis been performed appropriately and rigorously? 

Reviewer #1: N/A

4. Have the authors made all data underlying the findings in their manuscript fully available?

Reviewer #1: No

5. Is the manuscript presented in an intelligible fashion and written in standard English?

Reviewer #1: No

6. Review Comments to the Author

Reviewer #1: Reviewers’ Comments

Comments to the Author

Based on your response to my previous comments, I have further clarified the points that require additional explanation.

==

The Introduction Section:

1. Definition and Scope of Key Terms

“Mobility” and “housing” appear to be core terms in your study. However, the scope of these terms requires clarification. Additionally, the hypothesis mentioned on Page 5, Line 126, lacks specificity, as it only suggests that “novel insights” will be obtained. Consequently, the overall aim of the study remains unclear. Please elaborate on the definitions and scope of terms such as “mobility,” “housing,” and “housing-related decisions,” and discuss their significance to the study.

Your Response:

We revised the paragraph where “housing” is first introduced and provided a definition on Page 3, Lines 77–80:

“In our previous work, we found that housing decisions among older adults, i.e., the decision-making process of choosing whether to age at home or move into institutional residential care [5, 6], are the most prevalent among the difficult decisions they face [5, 7, 8].”

Reply:

It is stated, “In our previous work,” but was this research conducted by the same authors? The author names differ. Additionally, what does “the most prevalent” mean in this context? Please specify the data or studies supporting this claim and clarify the connection to the referenced previous work.

==

We also added a second paragraph on Page 3, Lines 81–86, contextualizing the importance of examining housing decisions and older adults’ experiences:

“Although there are many conceptual frameworks regarding housing decisions [9–13], social and emotional concerns of older adults, as well as their existing physical and social assets, are often overlooked [6, 14]. As a result, a housing decision may not fully align with their preferences, increasing their distress [5]. For example, an older adult with a habit of morning walks may relocate to an area where it is difficult to walk due to busy roads.”

Reply:

This paragraph highlights important context, but further elaboration is required to demonstrate how your study adds value. Please ensure the examples directly align with the study’s scope and clearly establish a gap in the literature.

==

Regarding “mobility,” we added a definition from a previous study by Meijering (2021) on Page 5, Lines 108–111:

“Mobility plays a key role in social health. While some definitions of mobility focus solely on physical ability, such as moving oneself from one place to another [24, 25], in this study, we consider that mobility also encompasses the purpose behind the movement, such as social interactions, family engagements, work, and leisure activities [26].”

Reply:

Your definition of mobility includes both physical and social dimensions. However, physical mobility tracked by GPS appears to be the primary variable in your study. How do you measure these broader aspects of mobility? Will interviews supplement GPS data? Please revise the Methods section to detail how you measure and interpret “multimodal mobility.”

==

We revised the hypothesis and clarified the overall goal of the study on Page 6, Lines 137–143:

“Following an earlier exploratory study [22, 41, 42], we hypothesized that mobility and social health are important dimensions in housing decisions. Therefore, with the overall goal of informing the design of decision aids, we aimed to map the activity spaces of older adults in two Canadian provinces. By combining GPS data with narrative sources (journals and interviews), we sought to highlight the importance of considering older adults’ existing social and physical assets in housing decisions.”

Reply:

The hypothesis emphasizes the importance of mobility and social health in housing decisions, but this seems self-evident. What is the novel contribution of your paper? Clearly articulate what remains unknown in this area and how your study addresses these gaps.

==

3. Citations and Academic Writing

On Page 5, Lines 120–122, a citation is attached to a question, which does not adhere to academic writing standards.

Your Response:

We revised the paragraph to remove the question format. The updated text appears on Pages 5–6, Lines 128–134:

“For instance, it is important to identify social supports, such as friends and family, that maintain an older adult’s social health [39]. Volunteering may provide a healthy sense of self-worth, enabled by their mobility [40]. Thus, it is crucial to understand the motivations that help them remain mobile and maintain physical health. If they decide to move, we must ask whether their social and physical assets will still be available to them.”

Reply:

In the sentence “If they decide to move elsewhere,” does “move” refer to relocating to a different home or neighborhood? Please clarify the intended meaning.

==

Conclusion Section:

Even if mobility patterns are associated with social health, how do they relate to housing transitions or decisions? The causal relationship is unclear.

Your Response:

We revised the Conclusion in the Abstract (Page 3, Lines 66–67):

“Mobility patterns and social assets are crucial for understanding older adults' social health and should be considered in tools designed to support housing decisions.”

Additionally, we rewrote the final sentences of the Conclusion section (Pages 34–35, Lines 770–775):

“These factors could inform housing decisions, which, in turn, could shape the design of decision aids. Incorporating these dimensions alongside conventional health-system considerations will contribute to decision-making tools that better align with older adults' needs.”

Reply:

These statements largely reiterate findings from previous studies. Summarize the relationship between mobility patterns, social health, and housing transitions based on existing literature. Return to the Introduction to establish this connection and clarify the novelty of your contribution.

Minor point

・Spelling error：Neighbourhood→Neighborhood

・Uploaded text is submitted before correction in review mode.

・Additional comments on Introduction

The following is your idea. Please revise the text by looking for findings from previous studies that support this idea.

Relevant text:

While healthcare professionals and family caregivers involved in housing decisions with older adults may be conscious of their growing health burdens and safety needs, they may be less aware of these existing social and physical assets and older adults’ access to them, which should play an equally important role in the housing decision.

7. PLOS authors have the option to publish the peer review history of their article (what does this mean? ). If published, this will include your full peer review and any attached files.

**Do you want your identity to be public for this peer review?** For information about this choice, including consent withdrawal, please see our Privacy Policy .

Reviewer #1: **Yes: ** Takeshi Endo

---

## [Author Response · Author response to Decision Letter 2]

27 Dec 2024

Academic Editor Comment :

The referee reviewed your revision and additionally raised some comments to the manuscript and recommended "Major Revision" again. I suggest the authors to follow the referee's comment and revise this manuscript carefully.

Response: Thank you for your feedback. We appreciate the opportunity to further refine our manuscript. We carefully addressed each of the referee's points and revised the manuscript accordingly.

Reviewer #1

Question 1=========================

1. Definition and Scope of Key Terms

“Mobility” and “housing” appear to be core terms in your study. However, the scope of these terms requires clarification. Additionally, the hypothesis mentioned on Page 5, Line 126, lacks specificity, as it only suggests that “novel insights” will be obtained. Consequently, the overall aim of the study remains unclear. Please elaborate on the definitions and scope of terms such as “mobility,” “housing,” and “housing-related decisions,” and discuss their significance to the study.

Your Response:

We revised the paragraph where “housing” is first introduced and provided a definition on Page 3, Lines 77–80:

“In our previous work, we found that housing decisions among older adults, i.e., the decision-making process of choosing whether to age at home or move into institutional residential care [5, 6], are the most prevalent among the difficult decisions they face [5, 7, 8].”

Reply:

It is stated, “In our previous work,” but was this research conducted by the same authors? The author names differ. Additionally, what does “the most prevalent” mean in this context? Please specify the data or studies supporting this claim and clarify the connection to the referenced previous work. First of all, thank you for giving us the opportunity to address and clarify these points. We appreciate this chance to make the manuscript clearer and more accessible for readers.

Response to Question 1======================

To clarify regarding the phrase "In our previous work," these studies were conducted by some members of the Canada Research Chair in Shared Decision Making and Knowledge Mobilization, the primary team responsible for Canadian data collection in this study as well. Regarding "the most prevalent," we meant the most commonly reported difficult decision for older adults, a finding that was confirmed in three studies made by this Canadian team.

However, we understood your point that the author list differs from some of the names mentioned in this article. As this article is part of a broader collaboration within a consortium that includes contributors from other countries, we have revised the paragraph as follows:

“A cross-sectional survey conducted with a pan-Canadian web-based panel of older adults aged 65 and over, assessed clinically significant decisional conflict (CSDC) [5]. The survey revealed that housing decisions – specifically, the decision-making process regarding whether to age at home or move into institutional residential care – are the most commonly reported difficult decisions faced by older adults. This finding was confirmed by studies among caregivers of older adults [6] and their home care providers [7].” Page 3, lines 77-82.

Question 2=========================

We also added a second paragraph on Page 3, Lines 81–86, contextualizing the importance of examining housing decisions and older adults’ experiences:

“Although there are many conceptual frameworks regarding housing decisions [9–13], social and emotional concerns of older adults, as well as their existing physical and social assets, are often overlooked [6, 14]. As a result, a housing decision may not fully align with their preferences, increasing their distress [5]. For example, an older adult with a habit of morning walks may relocate to an area where it is difficult to walk due to busy roads.”

Reply:

This paragraph highlights important context, but further elaboration is required to demonstrate how your study adds value. Please ensure the examples directly align with the study’s scope and clearly establish a gap in the literature.

Response to Question 2======================

Thank you for your suggestion. Below, we have revised this paragraph to highlights the gap in the literature and how our study addresses it, and adds an example directly aligned with our study:

“Studies show a strong relationship between wellbeing and social participation among older adults [8]. Other studies suggest that moving home may remove these social networks and other informal support systems [9]. One review of support for older adults making housing decisions suggests that such support is generally undermined by lack of attention to the whole person and lack of preparation for the move [10]. Secondly, while there are studies showing that mobility among older adults is important to a sense of autonomy and social engagement [11-14], few studies have explored mobility as a social asset in the context of older adults’ housing decisions [9, 15]. As an example, an older adult who is used to visiting nearby community centers may be relocated to an area with limited access to public transportation, thereby restricting their social interactions and undermining their social health. If tools that support housing decisions, such as decision aids, fail to take such considerations into account, increased decision regret is likely [16].” Pages 3 and 4, lines 83-94.

We have added this reference:

16. Diori HI. A Critical Insight into Needs Assessment Technique and the Way Social Needs are Actually Assessed. Advanced Journal of Social Science. 2021;8(1):3-9.

Question 3=========================

Regarding “mobility,” we added a definition from a previous study by Meijering (2021) on Page 5, Lines 108–111:

“Mobility plays a key role in social health. While some definitions of mobility focus solely on physical ability, such as moving oneself from one place to another [24, 25], in this study, we consider that mobility also encompasses the purpose behind the movement, such as social interactions, family engagements, work, and leisure activities [26].”

Reply:

Your definition of mobility includes both physical and social dimensions. However, physical mobility tracked by GPS appears to be the primary variable in your study. How do you measure these broader aspects of mobility? Will interviews supplement GPS data? Please revise the Methods section to detail how you measure and interpret “multimodal mobility.”

Response to Question 3======================

In our study, mobility is primarily measured by GPS, which we use to track movement and participants' travels, and complemented by daily journal entries. Our intention was to highlight that, although mobility is physical, there are factors influencing the purpose behind older adults' mobility. These influencing factors are assessed through interviews as part of our qualitative analysis.

We have revised the paragraph as follows:

“Mobility plays a key role in social health. Although the definition of mobility refers solely to the physical ability to move from one place to another [24, 25], there are influencing factors that shape the underlying purpose of this movement (e.g., social interactions, family engagements, work, and leisure activities) [26]. In this study, we consider mobility as a physical phenomenon, but also the influencing factors that determine its purpose.” Page 05, lines 116-120.

Regarding the methods, we described our study design as a convergent mixed-methods approach, integrating data in which one method complements the other. We have revised the paragraph explaining the combination of GPS data to measure mobility and qualitative data to capture the factors influencing movement.

“We chose this approach due to the building nature of the data [54], where the interviews deepened our understanding of the mobility captured by the GPS tracking, uncovering the factors influencing the mobility.” Page 10, lines 250-252.

Question 4=========================

We revised the hypothesis and clarified the overall goal of the study on Page 6, Lines 137–143:

“Following an earlier exploratory study [22, 41, 42], we hypothesized that mobility and social health are important dimensions in housing decisions. Therefore, with the overall goal of informing the design of decision aids, we aimed to map the activity spaces of older adults in two Canadian provinces. By combining GPS data with narrative sources (interviews), we sought to highlight the importance of considering older adults’ existing social and physical assets in housing decisions.”

Reply:

The hypothesis emphasizes the importance of mobility and social health in housing decisions, but this seems self-evident. What is the novel contribution of your paper? Clearly articulate what remains unknown in this area and how your study addresses these gaps. Thanks for the comment. We appreciate the opportunity to clarify the novel contributions of our study and how it addresses existing gaps in the literature.

Response to Question 4======================

While the importance of mobility and social health in housing decisions may seem self-evident at a conceptual level, at a practical level the tools that support decision making - rarely designed by older adults themselves - hardly consider these dimensions. Please see our new summary of the literature and gaps below (response to question 6).

In addition, in terms of methodology, few studies integrate objective mobility data (e.g. GPS tracking) with qualitative insights (e.g. interviews) to provide a comprehensive understanding of how older adults navigate their activity spaces and leverage their social and physical assets. Most studies focus on either social health or mobility in isolation, often only using self-reported measures that may not capture the complexity of these dimensions.

We have revised the manuscript, and the updated text now reads:

“Following earlier exploratory studies [22, 44, 45], we hypothesized that measuring the mobility patterns of older adults, in combination with qualitative interviews and journals regarding the social advantages that are associated with that mobility, would provide a portrait of their existing social and physical assets that constitutes essential information for their housing decisions. Existing research often isolates these factors [46-48] or relies on self-reported data [49-51]. Therefore, with the overall goal of informing the design of decision aids, we aimed to map the activity spaces of older adults in two Canadian provinces by combining GPS data with narrative sources (journals and interviews).” Page 6, lines 147-155.

To support this data, we added the following references:

46. Siren A, Hakamies-Blomqvist L. Mobility and Well-being in Old Age. Topics in Geriatric Rehabilitation. 2009;25(1):3-11.

47. Morris T, Manley D, Sabel CE. Residential mobility: Towards progress in mobility health research. Prog Hum Geogr. 2018;42(1):112-33.

48. Fuller-Thomson E, Hulchanski JD, Hwang S. The housing/health relationship: what do we know? Rev Environ Health. 2000;15(1-2):109-33.

49. Rantakokko M, Portegijs E, Viljanen A, Iwarsson S, Rantanen T. Life-space mobility and quality of life in community-dwelling older people. J Am Geriatr Soc. 2013;61(10):1830-2.

50. Rosenberg DE, Huang DL, Simonovich SD, Belza B. Outdoor built environment barriers and facilitators to activity among midlife and older adults with mobility disabilities. Gerontologist. 2013;53(2):268-79.

51. Ziegler F, & Schwanen, T. ‘I like to go out to be energised by different people’: An exploratory analysis of mobility and wellbeing in later life. Ageing and Society. 2011;31(15):758-81.

Question 5=========================

3. Citations and Academic Writing

On Page 5, Lines 120–122, a citation is attached to a question, which does not adhere to academic writing standards.

Your Response:

We revised the paragraph to remove the question format. The updated text appears on Pages 5–6, Lines 128–134:

“For instance, it is important to identify social supports, such as friends and family, that maintain an older adult’s social health [39]. Volunteering may provide a healthy sense of self-worth, enabled by their mobility [40]. Thus, it is crucial to understand the motivations that help them remain mobile and maintain physical health. If they decide to move, we must ask whether their social and physical assets will still be available to them.”

Reply:

In the sentence “If they decide to move elsewhere,” does “move” refer to relocating to a different home or neighborhood? Please clarify the intended meaning.

Response to Question 5======================

Thank you for your question. We have revised the sentence as you suggested:

"if they decide to move to a different home or neighborhood…” Page 6, lines 142-143.

Question 6=========================

Conclusion Section:

Even if mobility patterns are associated with social health, how do they relate to housing transitions or decisions? The causal relationship is unclear.

Your Response:

We revised the Conclusion in the Abstract (Page 3, Lines 66–67):

“Mobility patterns and social assets are crucial for understanding older adults' social health and should be considered in tools designed to support housing decisions.”

Additionally, we rewrote the final sentences of the Conclusion section (Pages 34–35, Lines 770–775):

“These factors could inform housing decisions, which, in turn, could shape the design of decision aids. Incorporating these dimensions alongside conventional health-system considerations will contribute to decision-making tools that better align with older adults' needs.”

Reply:

These statements largely reiterate findings from previous studies. Summarize the relationship between mobility patterns, social health, and housing transitions based on existing literature.

Response to Question 6======================

Return to the Introduction to establish this connection and clarify the novelty of your contribution. Thank you for pointing out the need to summarize the relationship between mobility patterns, social health, and housing transitions.

In fact, although there is some literature on mobility and social health among older adults there is very little on this relationship in the practical context of housing decisions. We improved the summary of existing literature in the introduction; and we revised the Conclusion to highlight the novelty of our study:

Introduction:

“Studies show a strong relationship between wellbeing and social participation among older adults [8]. Other studies suggest that moving home may remove these social networks and other informal support systems [9]. One review of support for older adults making housing decisions suggests that such support is generally undermined by lack of attention to the whole person and lack of preparation for the move [10]. Secondly, while there are studies showing that mobility among older adults is important to a sense of autonomy and social engagement [11-14], few studies have explored mobility as a social asset in the context of older adults’ housing decisions [9, 15]. As an example, an older adult who is used to visiting nearby community centers may be relocated to an area with limited access to public transportation, thereby restricting their social interactions and undermining their social health. If tools that support housing decisions, such as decision aids, fail to take such considerations into account, increased decision regret is likely [16].” Pages 3 and 4, lines 83-94.

Conclusion:

“This study underlines the importance of integrating mobility patterns and their associated social assets into housing decision frameworks. It provides guidance for the development of user-informed decision support tools, tailored specifically to Canadian older adults' housing decisions, that consider older adults’ exis

---

## [Decision Letter · Decision Letter 2]

30 Jan 2025

PONE-D-24-37265R2Exploring mobility patterns and social health of older Canadians living at home to inform decision aids about housing: a mixed-methods studyPLOS ONE

Dear Dr. Légaré,

Thank you for submitting your manuscript to PLOS ONE. After careful consideration, we feel that it has merit but does not fully meet PLOS ONE’s publication criteria as it currently stands. Therefore, we invite you to submit a revised version of the manuscript that addresses the points raised during the review process.

We look forward to receiving your revised manuscript.

Kind regards,

Li-Pang Chen

Academic Editor

PLOS ONE

Journal Requirements:

Additional Editor Comments:

The reviewer is happy with this version, but he/she further raises two minor issues. Hope that the authors can address it.

Reviewers' comments:

Reviewer's Responses to Questions

**Comments to the Author**

1. If the authors have adequately addressed your comments raised in a previous round of review and you feel that this manuscript is now acceptable for publication, you may indicate that here to bypass the “Comments to the Author” section, enter your conflict of interest statement in the “Confidential to Editor” section, and submit your "Accept" recommendation.

Reviewer #1: All comments have been addressed

2. Is the manuscript technically sound, and do the data support the conclusions?

Reviewer #1: Yes

3. Has the statistical analysis been performed appropriately and rigorously? 

Reviewer #1: Yes

4. Have the authors made all data underlying the findings in their manuscript fully available?

Reviewer #1: Yes

5. Is the manuscript presented in an intelligible fashion and written in standard English?

Reviewer #1: Yes

6. Review Comments to the Author

Reviewer #1: The authors responded politely to my concerns. Please supplement this with a description of the methodology in the abstract and the overall value of the study in the introduction.

Comment 1: Abstract

Applicability: Walking interviews and in-depth interviews provide insights into physical and social assets, as well as obstacles to social health and mobility.

　Comment:

　What kind of analysis did you use to analyze the qualitative and quantitative studies?

Comment2: Response to Question 2:

　“I see from your explanation that there is a strong relationship between wellbeing and social participation in older adults” and that ”social health may be impaired when social interaction is limited.” So far, this seems obvious. What new value does your study offer to what these previous studies have said?

7. PLOS authors have the option to publish the peer review history of their article (what does this mean? ). If published, this will include your full peer review and any attached files.

**Do you want your identity to be public for this peer review?** For information about this choice, including consent withdrawal, please see our Privacy Policy .

Reviewer #1: **Yes: ** Takeshi Endo

---

## [Author Response · Author response to Decision Letter 3]

14 Feb 2025

Academic Editor Comment: The reviewer is happy with this version, but he/she further raises two minor issues. Hope that the authors can address it.

Response: We appreciate the reviewer's positive feedback and will address the two minor issues raised to further improve the manuscript.

Reviewer #1

Question 1=========================

Comment 1: Abstract

Applicability: Walking interviews and in-depth interviews provide insights into physical and social assets, as well as obstacles to social health and mobility.

Comment:

What kind of analysis did you use to analyze the qualitative and quantitative studies?

Response to Question 1======================

First of all, thank you for considering our previous responses. We appreciate the opportunity to present these minor revisions.

We understand your point and we have added to the data analysis we performed for both qualitative and quantitative data to the Abstract, while keeping it within the 300-word limit. Pages 1 and 2, lines 39-66

“Introduction

Many tools support housing decisions for older adults but often overlook mobility patterns and social health. We explored these factors in older Canadians living at home to inform housing decisions.

Methods

We conducted a mixed-methods study with 20 older adults (65+) from Quebec and Alberta living independently or in senior residences with outdoor mobility. Data collection included sociodemographic information, GPS tracking, walking interviews, daily journals, and in-depth interviews. Data from interviews, which explored physical and social assets and barriers to social health and mobility, were analyzed using deductive content analysis in NVivo 12. GPS data were subjected to spatial analysis in QGIS (Quantum Geographic Information System) to map activity spaces and mobility patterns by the number and distance of activities, activity types, and modes of transportation. Daily journals were transcribed into an Excel spreadsheet and compared with GPS data. Overall analysis was guided hierarchically by qualitative data, utilizing verbatim narratives and visualization (activity space maps) to illustrate data convergence.

Results

Among 20 participants, 14 completed all activities, including GPS trackers. GPS maps showed participants mostly left home to drive for shopping or walking. Over 14 days, participants made an average of 10.4 (±5.8) trips and traveled 186.9 km (±130.4), averaging 16.8 km (±29.8) per day. Transportation modes included car (n=9), walking (n=5), and bus (n=2). Daily journals revealed that participants typically traveled alone. Interviews identified physical assets as libraries and supermarkets (n=10), while social assets were family support when desired (n=13) neighborhood familiarity (n=14), both contributing to social health. Winter weather was the most cited mobility barrier (n=13).

Conclusions

These findings provide actionable insights to guide the development of user-informed decision support tools tailored to the housing decisions of Canadian older adults.”

We have also added further details about our analysis in the body of the manuscript (Page 11, lines 262-268), as follows.

“Upon completion of data collection, interviews were transcribed, and data anonymized (manual encryption of personally identifiable information). A deductive content analysis was performed on the transcripts based on an adapted version of the Asset-Based Approach to Community Development (ABCD) framework [44, 64] using NVivo 12. Qualitative and quantitative data (socio-demographic, self-reported health and ability to perform daily tasks) were then organized into a descriptive summary. Statistical Analysis System (SAS) software was used to describe characteristics of participants.”

We have added the following reference:

64. Kretzmann JP, & McKnight, J. L. Building communities from the inside out: A path toward finding and mobilizing a community's assets. Chicago: ACTA Publications; 1993.

Question 2=========================

Comment2: Response to Question 2:

　“I see from your explanation that there is a strong relationship between wellbeing and social participation in older adults” and that ”social health may be impaired when social interaction is limited.”

So far, this seems obvious. What new value does your study offer to what these previous studies have said?

Response to Question 2======================

Thank you for your question.

We agree that our study confirms that “there is a strong relationship between wellbeing and social participation in older adults” and that “social health may be impaired when social interaction is limited,” as you say.

However, we very respectfully suggest that in our introduction we had further proposed that mobility has an important but little recognized relationship with social health, and it is this relationship, the focus of our study, that constitutes its originality. We have revised the sentences referring to mobility to highlight this:

Introduction (Page 4, lines 90-91 and 95):

“Few studies have explored the relationship between *mobility and social health* in the context of older adults’ housing decisions [9, 15] ... If tools that support housing decisions, such as decision aids, fail to take *mobility as a social asset* into account, increased decision regret is likely.”

We also made some small additions to the Discussion and Conclusions to remind the reader that our study is about social assets that are associated with mobility:

Discussion (Page 33, lines 728-730):

“These activities suggest a whole additional range of *mobility-related* social and physical assets of the older person in their current location that should be taken into account in any future design of decision aids about housing...”

Conclusion (Page 35, lines 782-784):

“...despite a decline in autonomy, numerous existing social and physical assets *associated with mobility* make important contributions to their social health...”

---

## [Decision Letter · Decision Letter 3]

26 Feb 2025

Exploring mobility patterns and social health of older Canadians living at home to inform decision aids about housing: a mixed-methods study

PONE-D-24-37265R3

Dear Dr. Légaré,

We’re pleased to inform you that your manuscript has been judged scientifically suitable for publication and will be formally accepted for publication once it meets all outstanding technical requirements.

Kind regards,

Li-Pang Chen

Academic Editor

PLOS ONE

Additional Editor Comments (optional):

The reviewer has recommended Acceptance of this manuscript.

Reviewers' comments:

Reviewer's Responses to Questions

**Comments to the Author**

1. If the authors have adequately addressed your comments raised in a previous round of review and you feel that this manuscript is now acceptable for publication, you may indicate that here to bypass the “Comments to the Author” section, enter your conflict of interest statement in the “Confidential to Editor” section, and submit your "Accept" recommendation.

Reviewer #1: All comments have been addressed

2. Is the manuscript technically sound, and do the data support the conclusions?

Reviewer #1: Yes

3. Has the statistical analysis been performed appropriately and rigorously? 

Reviewer #1: Yes

4. Have the authors made all data underlying the findings in their manuscript fully available?

Reviewer #1: Yes

5. Is the manuscript presented in an intelligible fashion and written in standard English?

Reviewer #1: Yes

6. Review Comments to the Author

Reviewer #1: The author replied to all my concerns. The author clarified areas of the study that were particularly unclear in the research methodology.

7. PLOS authors have the option to publish the peer review history of their article (what does this mean? ). If published, this will include your full peer review and any attached files.

**Do you want your identity to be public for this peer review?** For information about this choice, including consent withdrawal, please see our Privacy Policy .

Reviewer #1: **Yes: ** Takeshi Endo

---

## [Editor Report · Acceptance letter]

PONE-D-24-37265R3

PLOS ONE

Dear Dr. Légaré,

I'm pleased to inform you that your manuscript has been deemed suitable for publication in PLOS ONE. Congratulations! Your manuscript is now being handed over to our production team.

Kind regards,

on behalf of

Dr. Li-Pang Chen

Academic Editor

PLOS ONE